# TOAST 🍞: Transformer Optimization using Adaptive and Simple Transformations

## Abstract

Foundation models achieve State-of-the-art (SOTA) performance across different tasks, but their size and computational demands raise concerns about accessibility and sustainability. Existing efficiency methods often require additional retraining or fine-tuning, limiting their practicality. Recent findings suggest that deep neural networks exhibit internal representation similarities. While such similarities across different models have been exploited for enabling techniques such as model stitching and merging, intra-network redundancy remains underexplored as a source for efficiency gains. In this paper, we introduce Transformer Optimization using Adaptive and Simple Transformations (TOAST), a framework that exploits these redundancies to approximate entire transformer blocks with lightweight closed-form mappings, such as linear transformation or even the identity, without any additional training. Across SOTA pretrained vision models (e.g., ViT, DINOv2, DeiT) and datasets ranging from MNIST to ImageNet-1k, TOAST reduces parameters and computation while preserving, and in some cases improving, downstream performance. These results show that large portions of transformer depth can be replaced by trivial functions, opening a new perspective on efficient foundation models.

## 1 Introduction

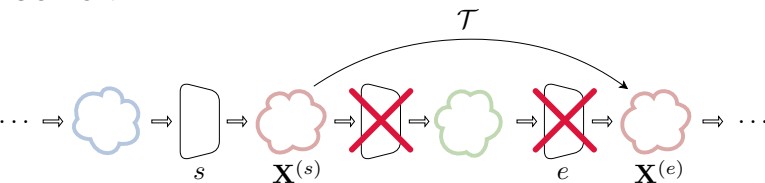

Figure 1: **Framework Description**. Given two latent spaces $\mathbf{X}^{(s)}$ and $\mathbf{X}^{(e)}$ corresponding to the outputs of blocks $s$ and $e$ for a random subset of 500 training samples, TOAST estimates a lightweight transformation $\mathcal{T}$ such that $\mathbf{X}^{(e)} \approx \mathcal{T}(\mathbf{X}^{(s)})$. This allows *entire* transformer blocks to be approximated by simple closed-form mappings (e.g., linear or identity), reducing parameters and computation without retraining.

As Neural Networks (NNs) continue to grow in size and complexity, their demand for computational resources has become a critical bottleneck. While larger models consistently achieve SOTA performance, this comes at the cost of substantial memory usage and power consumption, limiting their accessibility and deployment. This challenge is for instance most relevant in on-device scenarios, where saving memory, latency, and energy, even by little margins, is critical (Pan et al., 2022; Li et al., 2022). This has motivated a growing body of work on reducing model complexity. However, most existing approaches either require additional, resource-intensive training phases or lead to significant drops in accuracy. Recent studies reveal that there exists strong representational similarities both within and between NNs. In other words, when focusing on intra-network similarities, different blocks often perform overlapping functions or produce highly correlated outputs.

This redundancy suggests an opportunity: *instead of retraining or pruning, can we approximate these blocks with simpler transformations?* To address this question, we propose Transformer Optimization using Adaptive and Simple Transformations (TOAST), a novel framework that exploits block-level representational redundancy to replace transformer blocks with lightweight transformations. By

doing so, TOAST reduces parameter count and computational cost, while maintaining (and in some cases even improving) downstream task performance. Crucially, our method is training-free, making it simple, efficient, and widely applicable, even in resource-constrained scenarios such as deployment on edge devices, where even the smallest available models may exceed memory or power budgets. Our main contributions are as follows:

- We propose TOAST, a simple yet effective framework that replaces transformer blocks with lightweight transformations (e.g., linear maps or even the identity), significantly reducing parameters and computational cost while preserving downstream performance (Figure 1).

- We introduce linear approximation error as a stable and computationally lightweight criterion for identifying redundant transformer blocks (Tables 9 and 11 and Algorithms 1 and 2) and we present a systematic analysis of block-wise representational similarities in pre-trained vision transformers, revealing consistent redundancy patterns across diverse models and motivating the possibility of approximating entire blocks (Figures 2 and 6).

- We empirically demonstrate that accurate block approximations can be obtained from only a few hundred samples, showing that block redundancy can be exploited without requiring large-scale retraining (Tables 1 and 4 and Figure 5).

- We extensively validate our approach across a wide spectrum of vision models (e.g., `DiNO-B`, `ViT-L`, `DEiT-S`, `ViT-S`, `DiNO-S`, `ViT-T`) and datasets ranging from `MNIST` to `ImageNet1k`, confirming both the generality and efficiency of the method (Tables 1 to 3 and 12 to 16).

- We preliminarily validate the application of TOAST beyond vision classification, including semantic segmentation using `ViT-S` and `DiNO-B` on `SceneParse150`, and and text classification using `ModernBERT-B` on `AG News` (Tables 5 and 6 and Section A.2.4).

## 2 RELATED WORK

**Measuring Similarities** A range of metrics have been introduced to assess the similarity between latent spaces generated by different NNs (Klabunde et al., 2023; Ballester et al., 2023). One established approach is Canonical Correlation Analysis (CCA) (Hotelling, 1992), known for its invariance to linear transformations. Variants of CCA, such as Singular Value CCA (SVCCA) (Raghu et al., 2017), aim to enhance robustness, while techniques like Projection Weighted CCA (PWCCA) (Morcos et al., 2018) mitigate sensitivity to small perturbations. Another widely used metric, Centered Kernel Alignment (CKA) (Kornblith et al., 2019), captures the similarity between latent spaces while ignoring orthogonal transformations. However, recent work (Davari et al., 2022) highlights that this metric can be sensitive to shifts in the latent space. Additionally, Barannikov et al. (2021) proposes a method to compare two data representations by measuring the multi-scale topological dissimilarity, while Fumero et al. (2024) leverages the principles of spectral geometry to model and analyze the relationships between distinct latent spaces.

**Leveraging Similarities** Valeriani et al. (2024) examines the intrinsic dimensionality and neighbor compositions of representations in transformer models. Kvinge et al. (2022) investigates how models process variations in data points across layers, while Nguyen et al. (2020) assesses the impact of network depth and width on hidden representations. Additionally, Crisostomi et al. (2023) studies the conditions under which two latent spaces can be merged into a unified one. Moschella et al. (2023) constructs a unified space shared by different NNs, enabling zero-shot stitching of independently trained models across different modalities (Norelli et al., 2023). More recently, Cannistraci et al. (2024) enables model stitching without explicit assumptions about the transformation class connecting the latent manifold embeddings, or with only partial correspondence between latent spaces (Cannistraci et al., 2023). Finally, Lähner & Moeller (2024); Maiorca et al. (2024) demonstrate that representations learned by distinct NNs can be aligned using simple transformations.

**Architectural Efficiency** While large-scale models with billions or even trillions of parameters continue to achieve state-of-the-art performance, their growth comes with trade-offs, such as slower inference times and significantly higher computational costs. Improving the efficiency of Deep Neural Network (DNN) has been a long-standing area of research. For instance, Veit et al. (2016) shows that removing residual blocks from deep Convolutional Neural Networks (CNNs) only marginally

impacts performance, which inspired approaches to reduce inference time by dynamically deciding which layers to execute based on the input (Wu et al., 2018; Veit & Belongie, 2018). Additionally, various techniques to enhance efficiency have emerged, such as early exiting and model pruning. Early exit strategies, which introduce intermediate output layers at different stages of the network, have been shown to reduce inference time (Xin et al., 2020; Zhou et al., 2020; Yu et al., 2022; Tang et al., 2023). However, these approaches require the training of intermediate classifiers to enable exits at predefined layers. Alternatively, model pruning reduces computational load by either removing individual weights based on specific criteria, such as gradient information (Ma et al., 2023), entropy (Liao et al., 2023), or second-order information (Singh & Alistarh, 2020), or by eliminating larger structural components, like channels or residual blocks in ResNets (Bai et al., 2023; Wang & Wu, 2023), weights in LLMs (Sun et al., 2023) and self-attention layers in Transformers (Zhang & He, 2020; Sajjad et al., 2023; Venkataramanan et al., 2024; Zhang et al., 2024). Although effective, these approaches require training the model from scratch and, in the best case, fine-tuning. However, Bai et al. (2023) shows that for CNNs, this additional training step can sometimes be avoided.

Unlike other methods, TOAST leverages intra-network similarities to reduce vision transformers complexity *without the need for additional training steps* while maintaining competitive performance.

## 3 BLOCKS APPROXIMATION

The central idea of our approach is that it is possible to leverage representation similarities within transformer-based architectures to replace entire blocks with simpler transformations. In this work, a *block* refers to a sequence of layers including multi-head self-attention, normalization, and feed-forward layers, that function together as a cohesive unit. By replacing these blocks with simpler transformations, we can reduce the computational complexity of the network while maintaining its core functionality.

**Approximating Transformer Blocks** Given two blocks $s$ and $e$, our goal is to replace the intermediate blocks $s+1, \ldots, e$ with a single, lightweight transformation that maps the output of block $s$ directly to an approximation of the output of block $e$. This approach allows us to skip the computation of blocks $s+1, \ldots, e$, effectively reducing the overall computational costs. This approximation can be repeated for multiple, non-overlapping blocks, i.e., blocks $(s_i, e_i)$ and $(s_j, e_j)$ with $e_i < s_j$. An overview of the method is provided in Figure 1.

Let $\mathbf{X}^{(s)} \in \mathbb{R}^{|\mathcal{D}_{\text{sub}}| \times d_s}$ and $\mathbf{X}^{(e)} \in \mathbb{R}^{|\mathcal{D}_{\text{sub}}| \times d_e}$ represent the output representations from block $s$ and $e$ respectively, for the data points in $\mathcal{D}_{\text{sub}} \subset \mathcal{D}$, sampled uniformly at random from the full training dataset $\mathcal{D}$. Our objective is to find a transformation $\mathcal{T} : \mathbb{R}^{d_s} \to \mathbb{R}^{d_e}$ such that:

$$\mathbf{X}^{(e)} \approx \mathcal{T}(\mathbf{X}^{(s)})$$

In this work, we consider $\mathcal{T}$ to be the *identity* or a *linear transformation* $\mathbf{T}$. We can compute the linear transformation $\mathbf{T}$ by minimizing the squared error between the transformed output $\mathcal{T}(\mathbf{X}^{(s)})$ and the actual $\mathbf{X}^{(e)}$:

$$\mathbf{T} = \arg\min_{\mathcal{T}} \|\mathbf{X}^{(e)} - \mathcal{T}(\mathbf{X}^{(s)})\|_2^2$$

This optimization problem allows for a closed-form solution that efficiently computes the optimal transformation $\mathbf{T}$. The solution bypasses the computation of *all* layers between any two blocks $s$ and $e$, replacing them with $\mathbf{T}$. This approximation reduces computational complexity while minimally affecting internal representations, as illustrated in Figures 7 to 11, and preserves compatibility with downstream classifiers, achieving significant compression as shown in Tables 1 to 3 and 12 to 15.

**Patterns of Similarity between Transformer Blocks** Inspired by existing results Venkataramanan et al. (2024), which show that multi-head attention modules exhibit similarity in learned representations, we investigate whether pre-trained foundation models contain *entire blocks* that produce highly similar representations. Rather than using CKA to measure representational similarity, we quantify how well the output of a later block can be reconstructed from an earlier one using a simple linear transformation. All representations are computed using only the `[CLS]` token, providing a consistent and semantically aligned basis for comparing blocks.

Given representations $\mathbf{H}_s$ and $\mathbf{H}_e$ extracted from blocks $s < e$, we learn the optimal linear map $\mathbf{W}^*$ that solves

$$\mathbf{W}^* = \arg\min_{\mathbf{W}} \|\mathbf{H}_e - \mathbf{H}_s\mathbf{W}\|_F^2.$$

We measure similarity via the normalized residual error

$$\epsilon(s, e) = \frac{\|\mathbf{H}_e - \mathbf{H}_s\mathbf{W}^*\|_F}{\|\mathbf{H}_e\|_F},$$

where lower values indicate that block $e$'s representations are well explained by a linear transformation of block $s$.

By computing the metric for all block pairs, using only a small random subset of the training data (i.e., 50 samples), and ranking them, we can automatically identify blocks whose computations contribute minimally beyond a near-linear mapping. We additionally perform an ablation study comparing several candidate similarity metrics for block selection, and we report these results in Section A.1.3. The procedure used to automatically extract the top-$k$ skip candidates is summarized in Algorithm 1, and the linear approximation error is detailed in Algorithm 2.

## 4 EXPERIMENTS

In this section, we first analyze the similarities between different transformer blocks to motivate their approximation using simple transformations. We then present comprehensive results on image classification across various models and datasets to demonstrate the effectiveness and efficiency of the proposed method. Beyond these core results, we further study the robustness of TOAST through ablations on the number of samples required for approximation and the choice of translator architecture. Overall, our findings show that TOAST achieves strong performance while producing lighter and faster models. Due to space constraints, additional results on zero-shot image classification, as well as further qualitative and quantitative analyses, are provided in the Appendix (Sections A.2.2 and A.2.3).

### 4.1 LATENT ANALYSIS

In this section we investigate similarities in the latent representations of `DiNO-B` and `DEiT-S` on five datasets: `CIFAR-10`, `CIFAR-100`, `MNIST`, `F-MNIST`, and `ImageNet1k`. We compute the linear approximation error using only the `[CLS]` token, averaged over a small subset of 50 training samples. This is sufficient to reveal block-level similarity patterns while remaining computationally efficient. Additional results with other pretrained vision transformers (`ViT-T`, `ViT-S`, `DiNO-S`, `ViT-B`) are provided in Section A.2.1, showing consistent patterns for each model across different datasets.

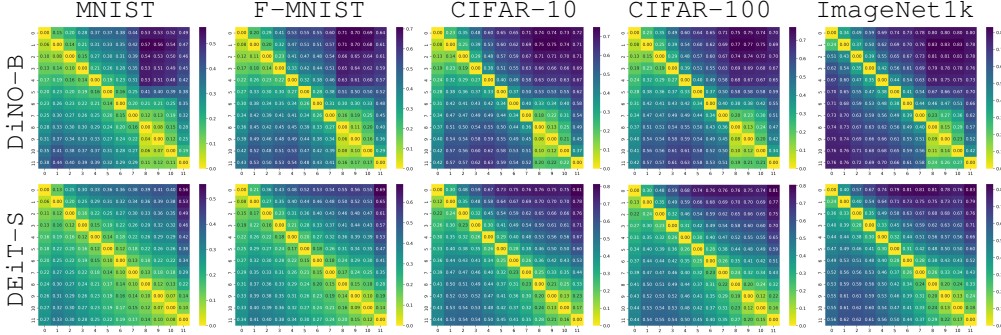

Figure 2: **Block Similarities**. Block-by-block similarities in `DiNO-B`, and `DEiT-S` models across five datasets: `MNIST`, `F-MNIST`, `CIFAR-10`, `CIFAR-100` and `ImageNet1k`. Each matrix quantifies the linear approximation error using only the `[CLS]` token, averaged over a small subset of 50 training samples. The matrices reveal that the similarity between blocks is predominantly influenced by the model rather than the specific dataset. Additional results in Section A.2.1.

**Do vision transformer models exhibit block-wise similarity patterns?**   The results in Figure 2 reveal that while the similarity patterns differ across models, they remain largely consistent for the same model across different datasets. This suggests that the similarity structure between computational blocks is predominantly influenced by the model itself. Although the general similarity pattern remains the same, the differences in values become more pronounced (i.e., the block structure becomes more evident) as the complexity of the dataset increases (e.g., from MNIST to ImageNet1k). These finding aligns with observations from Nguyen et al. (2020), where DNN trained from scratch exhibit a distinctive "block structure" in their representations, which is linked to model overparameterization. Our results extend this observation to vision pre-trained foundation models, showing that such a structure is primarily an intrinsic property of the model. Moreover, these consistent block-wise patterns indicate potential targets for approximation, suggesting that entire blocks may be replaced with simpler transformations without substantially altering the model's internal representations.

> **Takeaway** Pre-trained vision foundation models present block-wise similarity patterns that are primarily determined by the model itself.

**How does TOAST affect latent representations?**   We next analyze the impact of the proposed transformations on the final block's latent representations, which are used for downstream classification. We approximate these blocks using a shared linear transformation applied across all tokens, estimated on a subset of 500 training samples. For consistency, we use the same models and datasets as in Figure 2. To quantify the effect of the approximation, following (Venkataramanan et al., 2024; Kornblith et al., 2019) we compute the CKA similarity between the final block representations of the original and the TOAST-approximated model for each block $k$ using its preceding block as input. As shown in Figure 3, the model-specific similarity patterns re-emerge after approximation. The plots highlight more specific trends. Approximating blocks is easier on simpler tasks (e.g., image classification on MNIST or F-MNIST), yielding representations that closely match the originals, whereas on more complex datasets (e.g., ImageNet1k or CIFAR-100), the approximated representations deviate more from the original ones. Furthermore, the final blocks of DEiT-S exhibit high similarity, suggesting that approximating these layers preserves the final representations, while earlier blocks remain more critical. To provide a more intuitive view, Figure 4 visualizes the final-layer representations using Principal Component Analysis (PCA). We compare the original representations with those obtained after approximating the final block (10 → 11) using TOAST on F-MNIST, with colors indicating the 10 classes. The visualization confirms that approximating the final block of DiNO-B results in noticeable deviations from the original representations, whereas for DEiT-S the approximated representations remain highly similar.

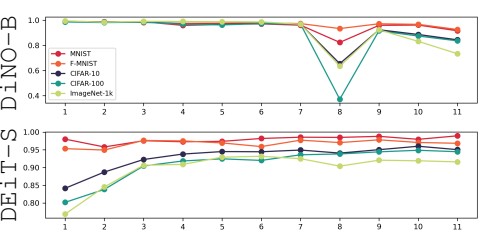

Figure 3: **Approximation vs. Representation Similarity**. CKA between the last block representations of the original and the approximated model when approximating the $i^{\text{th}}$ block.

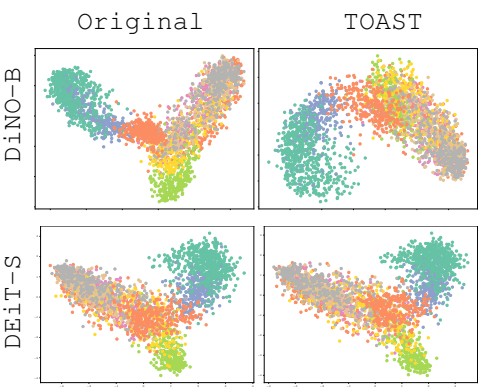

Figure 4: **PCA Visualization**. Final block representations for original and TOAST models on F-MNIST reveal DiNO-B's stronger reliance on final block compared to DEiT-S.

These observations align with the CKA analysis in Figure 3, highlighting that the effect of block approximation depends strongly on the model and its internal block structure. Additional results across other models and datasets are provided in Section A.2.1.

**Takeaway** Transformer blocks can be approximated using simple transformations, without compromising representation fidelity.

**Can entire transformer blocks be approximated without losing accuracy?** Initial results, reported in Table 1, support the qualitative analysis and empirically demonstrate that *entire vision transformer blocks* can be effectively approximated using simpler transformations (e.g., linear projections or, in some cases, the identity function). Such approximations reduce both the number of parameters and Giga Floating-Point Operations (GFLOPs), thereby improving throughput (images per second), while incurring only a slight to negligible decrease in downstream task performance. For instance, consistent with our earlier analysis, we find that approximating the final block of `DEiT-S` when using `ImageNet1k` (e.g., approximating blocks $10 \rightarrow 11$ or $9 \rightarrow 11$ with a linear transformation) yields modest performance drops going from 73.85% to 73.78% and 70.01%, respectively, while providing substantial efficiency gains. Importantly, we also show that even the identity transformation, achieves competitive results, with accuracy drops as small as -0.24% and -5.44%, respectively. However, the choice of translator naturally depends on the efficiency-accuracy trade-off: linear translation guarantee in general most reliable accuracy–efficiency balance, whereas the identity yields the leanest training-free approximation when maximum simplicity is required. Further methodological details and the full evaluation are presented in Section A.1 and Section 4.2 respectively, while details on the efficiency metrics and additional analysis on those are in Sections A.1.7 and A.2.6, respectively.

Table 1: **TOAST Image Classification Performance**. Performance comparison using the Identity translator and the Linear Translator for `DEiT-S` and `ImageNet1k` accross 3 seeds. The "Approx." column specifies the blocks used for approximation, the first one represents the block whose output is used to approximate the second block's output. Additional results in Tables 2 and 3 and Section A.2.2.

| Approx. | Params. | Identity Translator | | | Linear Translator | | |
|---|---|---|---|---|---|---|---|
| | | Accuracy % ↑ | GFLOPS ↓ | imgs/s ↑ | Accuracy % ↑ | GFLOPS ↓ | imgs/s ↑ |
| $2 \rightarrow 4$ | $-3.25$M | $63.74 \pm 0.19(-13.69\%)$ | 4.15 | 7222.5 | $69.87 \pm 0.14(-5.39\%)$ | 4.18 | 7187.6 |
| $9 \rightarrow 11$ | $-3.25$M | $69.83 \pm 0.33(-5.44\%)$ | 4.15 | 7224.6 | $70.01 \pm 0.27(-5.20\%)$ | 4.18 | 7203.8 |
| $0 \rightarrow 1$ | $-1.62$M | $64.02 \pm 0.08(-13.31\%)$ | 4.56 | 6755.8 | $62.32 \pm 0.15(-15.61\%)$ | 4.59 | 6748.9 |
| $10 \rightarrow 11$ | $-1.62$M | $\mathbf{73.67 \pm 0.26(-0.24\%)}$ | 4.56 | 6751.7 | $\mathbf{73.78 \pm 0.28(-0.10\%)}$ | 4.59 | 6756.3 |
| original | $21.81$M | $73.85 \pm 0.39$ | 4.97 | 6349.2 | $73.85 \pm 0.39$ | 4.97 | 6325.6 |

**Takeaway** TOAST effectively reduces model parameters and improve model efficiency without significantly compromising the downstream task performance.

## 4.2 Image Classification Performance

We evaluate TOAST on image classification tasks using pretrained models of varying sizes (`ViT-L`, `DiNO-B`, and `DEiT-S`) and two benchmark datasets (`CIFAR-100F` and `ImageNet1k`). Additional results with a broader set of models (`ViT-T`, `ViT-S`, `ViT-B`, `ViT-L`, `DiNO-S`, `DiNO-B`, `DEiT-S`) and datasets (`MNIST`, `F-MNIST`, `CIFAR-10`, `CIFAR-100C`) are provided in Section A.2.2. While in Section A.2.7, we complement the quantitative evaluations with qualitative analyses of misclassifications after block approximation, providing further insight into model behavior under TOAST. Additional implementation details, including model and dataset specifications, computational resources, and software tools, are provided in Tables 7 and 8, and Sections A.1.5 to A.1.7.

Block approximations in TOAST are calculated via a shared linear, or identity, transformation applied across all tokens and are estimated using a subset of 500 training samples. A linear classifier is then trained on top of the frozen backbone with the Adam optimizer (learning rate 0.001), batch size 256, for 5 epochs, over 3 different seeds. This setup simulates a realistic scenario where a pretrained feature extractor is adapted to a new dataset unseen during pretraining. However, to assess the robustness of our method, we also report the results using the original classification heads (Section A.2.5), which confirm the consistency of our findings.

**Are TOAST results competitive?** As shown in Tables 2 and 3, TOAST consistently reduces model size and GFLOPs while maintaining, and in some cases improving, image classification

Table 2: **TOAST Classification Performance on `ImageNet1k`**. Image classification accuracy, GFLOPs, and throughput for `DEiT-S`, `DiNO-B`, and `ViT-L` using `ImageNet1k`. The "Approx." column indicates the block pairs where the first block approximates the second. Additional results using other models and datasets are provided in Table 3 and Section A.2.2.

| | Approx. | Params. | Identity | | | Linear | | |
|---|---|---|---|---|---|---|---|---|
| | | | Accuracy % ↑ | GFLOPs ↓ | imgs/s ↑ | Accuracy % ↑ | GFLOPs ↓ | imgs/s ↑ |
| DEiT-S | $3 \to 4, 9 \to 11$ | -4.88M | $66.96 \pm 0.34(-9.33\%)$ | 3.74 | 7751.4 | $68.39 \pm 0.13(-7.39\%)$ | 3.80 | 7718.4 |
| | $3 \to 4, 9 \to 10$ | -3.25M | $69.22 \pm 0.13(-6.27\%)$ | 4.15 | 7210.9 | $71.35 \pm 0.22(-3.38\%)$ | 4.21 | 7188.4 |
| | $2 \to 3$ | -1.62M | $70.80 \pm 0.05(-4.12\%)$ | 4.56 | 6754.2 | $73.19 \pm 0.19(-0.88\%)$ | 4.59 | 6736.7 |
| | $10 \to 11$ | -1.62M | $\mathbf{73.67 \pm 0.26}(-\mathbf{0.24\%})$ | 4.56 | 6752.6 | $\mathbf{73.78 \pm 0.28}(-\mathbf{0.09\%})$ | 4.59 | 6740.5 |
| | original | 21.81M | $73.85 \pm 0.39$ | 4.97 | 6349.2 | $73.85 \pm 0.39$ | 4.97 | 6325.6 |
| DiNO-B | $0 \to 4$ | -26.00M | $3.58 \pm 0.06(-95.20\%)$ | 16.32 | 3230.9 | $27.70 \pm 0.19(-62.71\%)$ | 16.47 | 3227.7 |
| | $0 \to 1, 2 \to 3, 4 \to 5$ | -19.50M | $6.98 \pm 0.18(-90.63\%)$ | 18.34 | 2947.0 | $61.02 \pm 0.36(-17.86\%)$ | 18.80 | 2929.6 |
| | $0 \to 1, 2 \to 3$ | -13.00M | $13.28 \pm 0.46(-82.18\%)$ | 20.37 | 2703.9 | $70.82 \pm 0.49(-4.66\%)$ | 20.67 | 2681.2 |
| | $0 \to 1$ | -6.50M | $\mathbf{65.47 \pm 0.43}(-\mathbf{12.14\%})$ | 22.39 | 2506.6 | $\mathbf{73.43 \pm 0.02}(-\mathbf{1.15\%})$ | 22.54 | 2487.0 |
| | $5 \to 6$ | -6.50M | $28.84 \pm 0.51(-61.30\%)$ | 22.39 | 2503.1 | $73.01 \pm 0.41(-1.71\%)$ | 22.54 | 2490.6 |
| | original | 86.58M | $74.52 \pm 0.26$ | 24.42 | 2321.3 | $74.52 \pm 0.26$ | 24.42 | 2316.5 |
| ViT-L | $2 \to 4, 18 \to 23$ | -80.83M | $62.92 \pm 0.21(-19.89\%)$ | 45.05 | 1654.9 | $67.43 \pm 0.05(-14.16\%)$ | 45.47 | 1652.8 |
| | $17 \to 23$ | -69.28M | $66.81 \pm 0.34(-14.95\%)$ | 47.70 | 1572.4 | $66.87 \pm 0.52(-14.87\%)$ | 47.90 | 1567.0 |
| | $3 \to 4, 19 \to 23$ | -57.74M | $70.97 \pm 0.42(-9.65\%)$ | 50.34 | 1509.9 | $71.50 \pm 0.14(-8.98\%)$ | 50.75 | 1499.5 |
| | $3 \to 4, 20 \to 23$ | -46.19M | $73.49 \pm 0.18(-6.44\%)$ | 52.98 | 1440.4 | $74.03 \pm 0.43(-5.76\%)$ | 53.39 | 1436.8 |
| | $3 \to 4, 21 \to 23$ | -34.64M | $75.80 \pm 0.26(-3.50\%)$ | 55.62 | 1377.2 | $76.30 \pm 0.14(-2.86\%)$ | 56.03 | 1345.6 |
| | $7 \to 8, 15 \to 16$ | -23.09M | $76.81 \pm 0.28(-2.21\%)$ | 58.26 | 1318.2 | $77.32 \pm 0.48(-1.56\%)$ | 58.67 | 1316.4 |
| | $16 \to 17, 22 \to 23$ | -23.09M | $77.64 \pm 0.32(-1.15\%)$ | 58.26 | 1318.8 | $77.64 \pm 0.02(-1.16\%)$ | 58.67 | 1312.3 |
| | $3 \to 4$ | -11.55M | $77.32 \pm 0.29(-1.57\%)$ | 60.90 | 1269.2 | $\mathbf{78.36 \pm 0.26}(-\mathbf{0.24\%})$ | 61.11 | 1270.0 |
| | $22 \to 23$ | -11.55M | $\mathbf{78.32 \pm 0.09}(-\mathbf{0.29\%})$ | 60.90 | 1267.5 | $78.21 \pm 0.19(-0.43\%)$ | 61.11 | 1270.9 |
| | original | 304.35M | $78.55 \pm 0.20$ | 63.54 | 1219.8 | $78.55 \pm 0.20$ | 63.54 | 1225.2 |

accuracy. This aligns with our representational analyses in Section 4.1: for instance, approximating the final block of `DEiT-S` produces latent representations nearly identical to the original (Figures 3 and 4), making it an ideal candidate for approximation. Even when multiple consecutive blocks are approximated (e.g., 9→11), models maintain performance comparable to or exceeding the original while significantly reducing parameters. This demonstrates that a simple linear transformation, or even the identity in certain cases, is sufficient to capture the functionality of full transformer blocks without additional training, provided the transformation is shared across all tokens.

Table 3: **TOAST Classification Performance on `CIFAR-100F`**. Image classification accuracy, GFLOPs, and throughput for `DEiT-S`, `DiNO-B`, and `ViT-L` using `CIFAR-100F`. The "Approx." column indicates the block pairs where the first block approximates the second. Additional results using other models and datasets are provided in Section A.2.2.

| | Approx. | Params. | Identity | | | Linear | | |
|---|---|---|---|---|---|---|---|---|
| | | | Accuracy % ↑ | GFLOPs ↓ | imgs/s ↑ | Accuracy % ↑ | GFLOPs ↓ | imgs/s ↑ |
| DEiT-S | $3 \to 4, 9 \to 11$ | -4.88M | $68.48 \pm 0.34(-3.44\%)$ | 3.74 | 7755.1 | $70.64 \pm 0.37(-0.39\%)$ | 3.80 | 7713.7 |
| | $9 \to 11$ | -3.25M | $\mathbf{72.28 \pm 0.36}(+\mathbf{1.92\%})$ | 4.15 | 7226.6 | $\mathbf{72.04 \pm 0.42}(+\mathbf{1.57\%})$ | 4.18 | 6791.7 |
| | $8 \to 9$ | -1.62M | $71.34 \pm 0.10(+0.60\%)$ | 4.56 | 6755.2 | $70.80 \pm 0.12(-0.17\%)$ | 4.59 | 6739.9 |
| | $9 \to 10$ | -1.62M | $71.66 \pm 0.39(+1.04\%)$ | 4.56 | 6692.1 | $71.49 \pm 0.20(+0.80\%)$ | 4.59 | 6741.3 |
| | original | 21.81M | $70.92 \pm 0.18$ | 4.97 | 6349.0 | $70.92 \pm 0.18$ | 4.97 | 6249.4 |
| DiNO-B | $0 \to 4$ | -26.00M | $18.29 \pm 0.86(-79.09\%)$ | 16.32 | 3233.8 | $62.25 \pm 0.54(-28.83\%)$ | 16.47 | 3204.9 |
| | $0 \to 1, 2 \to 3, 4 \to 5$ | -19.50M | $29.05 \pm 0.31(-66.79\%)$ | 18.34 | 2943.1 | $79.06 \pm 0.27(-9.60\%)$ | 18.80 | 2922.6 |
| | $0 \to 1, 2 \to 3$ | -13.00M | $33.25 \pm 0.18(-61.99\%)$ | 20.37 | 2705.6 | $84.18 \pm 0.18(-3.76\%)$ | 20.67 | 2690.1 |
| | $0 \to 1$ | -6.50M | $\mathbf{78.83 \pm 0.22}(-\mathbf{9.87\%})$ | 22.39 | 2492.8 | $\mathbf{86.64 \pm 0.37}(-\mathbf{0.94\%})$ | 22.54 | 2493.8 |
| | $2 \to 3$ | -6.50M | $47.51 \pm 0.52(-45.68\%)$ | 22.39 | 2484.2 | $86.06 \pm 0.20(-1.60\%)$ | 22.54 | 2484.6 |
| | original | 86.58M | $87.46 \pm 0.04$ | 24.42 | 2315.5 | $87.46 \pm 0.04$ | 24.42 | 2317.3 |
| ViT-L | $2 \to 4, 18 \to 23$ | -80.83M | $74.41 \pm 0.44(-13.79\%)$ | 45.05 | 1655.7 | $84.02 \pm 0.39(-2.66\%)$ | 45.47 | 1649.6 |
| | $17 \to 23$ | -69.28M | $85.32 \pm 0.45(-1.16\%)$ | 47.69 | 1578.8 | $84.55 \pm 0.44(-2.05\%)$ | 47.90 | 1552.1 |
| | $3 \to 4, 19 \to 23$ | -57.74M | $84.23 \pm 0.08(-2.43\%)$ | 50.34 | 1503.6 | $85.81 \pm 0.39(-0.59\%)$ | 50.75 | 1497.4 |
| | $3 \to 4, 20 \to 23$ | -46.19M | $84.68 \pm 0.18(-1.90\%)$ | 52.98 | 1445.2 | $86.30 \pm 0.11(-0.03\%)$ | 53.39 | 1431.0 |
| | $20 \to 23$ | -34.64M | $\mathbf{86.61 \pm 0.07}(+\mathbf{0.33\%})$ | 55.62 | 1381.2 | $86.55 \pm 0.22(+0.27\%)$ | 55.82 | 1372.6 |
| | $3 \to 4, 21 \to 23$ | -34.64M | $84.86 \pm 0.28(-1.70\%)$ | 55.62 | 1376.7 | $86.37 \pm 0.28(+0.06\%)$ | 56.03 | 1372.7 |
| | $20 \to 22$ | -23.09M | $86.30 \pm 0.23(-0.03\%)$ | 58.26 | 1317.5 | $86.52 \pm 0.12(+0.24\%)$ | 58.47 | 1314.6 |
| | $3 \to 4, 21 \to 22$ | -23.09M | $84.58 \pm 0.19(-2.02\%)$ | 58.26 | 1315.8 | $86.20 \pm 0.11(-0.14\%)$ | 58.67 | 1317.6 |
| | $20 \to 21$ | -11.55M | $86.44 \pm 0.24(+0.14\%)$ | 60.90 | 1268.5 | $86.39 \pm 0.08(+0.08\%)$ | 61.11 | 1266.7 |
| | $21 \to 22$ | -11.55M | $86.55 \pm 0.01(+0.26\%)$ | 60.90 | 1270.7 | $\mathbf{86.72 \pm 0.24}(+\mathbf{0.46\%})$ | 61.11 | 1269.2 |
| | original | 304.35M | $86.32 \pm 0.08$ | 63.54 | 1223.1 | $86.32 \pm 0.08$ | 63.54 | 1224.3 |

Additionally, efficiency gains are notable: throughput (imgs/s) increases while GFLOPs decreases, highlighting practical benefits for deployment, as also shown in Section A.2.6. Additional results

across other models (`DiNO-B`, `ViT-L`) and datasets confirm that TOAST generalizes across architectures and scales (Section A.2.2). Finally, while approximations are easier for simpler datasets (e.g., `CIFAR-100F`), TOAST still achieves meaningful compression with minimal accuracy loss on complex datasets like `ImageNet1k`. Additional results across models and datasets are provided in Tables 12 to 15. To assess scalability, we applied TOAST to `ViT-L`. Approximating selected blocks, e.g., $17 \rightarrow 23$, reduces the parameter count by 69.3M, lowers GFLOPs from 63.54 to 47.79, and increases throughput from 1223.1 to 1578.8 imgs/s, while incurring a minimal accuracy drop of 1.16%. This shows TOAST's utility in balancing substantial computational savings with a modest performance trade-off, even in large models.

> **Takeaway** Approximating selected blocks enables efficiency gains with minimal impact on the accuracy.

**Are 500 training samples enough?** We study the sensitivity of block approximation to the number of training samples using `DiNO-B` and `DEiT-S` on `ImageNet1k`. As shown in Figure 5, performance typically plateaus quickly: 500 samples are sufficient to obtain stable and reliable approximations. Increasing the sample count beyond this threshold provides only marginal gains, while substantially fewer samples lead to noticeable degradation. Interestingly, when the representational spaces of consecutive blocks are already highly aligned, even as few as 10 or 50 samples suffice to achieve competitive approximations. Conversely, for blocks that are harder to approximate, such as the early layers of `DEiT-S` (e.g., $0 \rightarrow 1$), even 4000 samples are insufficient to estimate a linear transformation that maintains competitive performance. We highlight that these results are obtained on `ImageNet1k`, which contains 1000 classes. The 500 samples represent only a small subset of the class space, yet reliable approximations are still achieved. This indicates that TOAST primarily captures the block-level structure of representations rather than requiring exhaustive coverage of all classes. Consequently, TOAST could be practical also in scenarios where a large labeled datasets is limited.

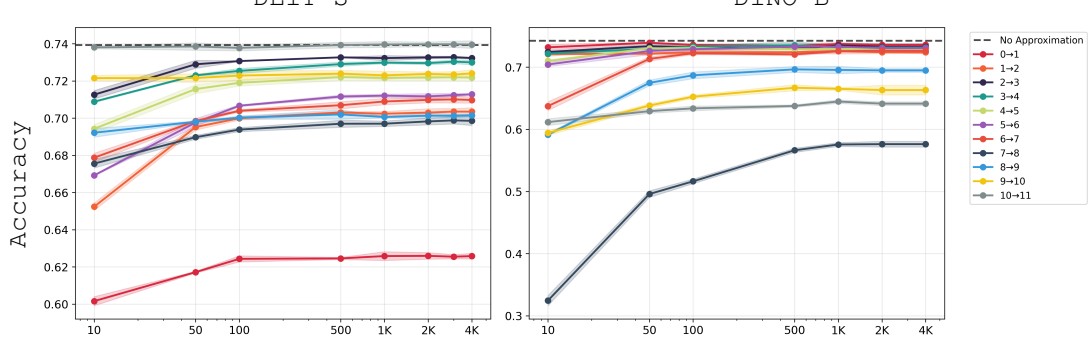

Figure 5: **Sample Size Ablation**. Classification accuracy as a function of the number of training samples used for approximating different layers of `DiNO-B` and `DEiT-S` with a linear transformation using `ImageNet1k`. Accuracy stabilizes after approximately 500 samples.

> **Takeaway** A small number of samples is sufficient to achieve stable and reliable representations when approximating transformer blocks, balancing efficiency and accuracy.

**What if a more complex transformation is used?** We evaluate whether deeper approximators improve downstream task performance. Specifically, we compare TOAST (Identity and Linear) to MultiLayer Perceptron (MLP) and Residual MLP, trained for 300 steps with Adam (learning rate $10^{-3}$). These more complex transformation, as for Identity and Linear, are applied across all tokens, and estimated using a subset of 500 training samples. Results in Table 4 show a consistent trend for `ViT-L` on both `ImageNet1k` and `CIFAR-100F`: the linear transformation provides the most reliable trade-off across datasets. On `CIFAR-100F`, linear often achieves the best or near-best accuracy (e.g., $21 \rightarrow 22$: 86.72% vs. 86.82% for Res-MLP and 85.20% for MLP), while remaining training-free, thus more efficient. On `ImageNet1k`, the gap becomes even clearer: for the same blocks linear reaches 77.24%, while Res-MLP and MLP reach 77.14% and 74.20%, respectively.

Additionally, also Linear obtain competitive results. TOAST operates in closed form, requires no optimization, and consistently achieves strong efficiency–accuracy trade-offs. These findings confirm that a simple linear transformation is sufficient to approximate transformer blocks in most settings, with deeper translators offering little benefit despite their higher cost.

Table 4: **Transformations Comparison**. Classification accuracy on `CIFAR-100F` and `ImageNet1k` using `ViT-L`. The "Approx." column specifies the block mapping (output of the first block is used to approximate the output of the second). MLP and Res-MLP are trained approximators, while Identity and Linear are closed-form and training-free. Results are averaged over three seeds.

| | Approx. | Params. | Accuracy ↑ | | | |
| | | | Identity | Linear | MLP | Res-MLP |
|---|---|---|---|---|---|---|
| CIFAR-100F | $3 \to 4, 20 \to 23$ | -46.19M | $84.68 \pm 0.18$ | $86.30 \pm 0.11$ | $84.36 \pm 0.48$ | $86.10 \pm 0.39$ |
| | $17 \to 23$ | -69.28M | $84.58 \pm 0.19$ | $86.20 \pm 0.11$ | $84.83 \pm 0.31$ | $86.49 \pm 0.08$ |
| | $3 \to 4, 19 \to 23$ | -57.74M | $84.23 \pm 0.08$ | $85.81 \pm 0.39$ | $83.63 \pm 0.42$ | $85.58 \pm 0.06$ |
| | $20 \to 23$ | -34.64M | $86.61 \pm 0.07$ | $86.55 \pm 0.22$ | $84.68 \pm 0.39$ | $86.19 \pm 0.02$ |
| | $3 \to 4, 21 \to 23$ | -34.64M | $84.86 \pm 0.28$ | $86.37 \pm 0.28$ | $84.90 \pm 0.71$ | $86.10 \pm 0.37$ |
| | $20 \to 22$ | -23.09M | $86.30 \pm 0.23$ | $86.52 \pm 0.12$ | $84.97 \pm 0.18$ | $86.71 \pm 0.28$ |
| | $3 \to 4, 21 \to 22$ | -23.09M | $84.58 \pm 0.19$ | $86.20 \pm 0.11$ | $84.83 \pm 0.31$ | $86.49 \pm 0.08$ |
| | $20 \to 21$ | -11.55M | $86.44 \pm 0.24$ | $86.39 \pm 0.08$ | $84.40 \pm 0.70$ | $86.63 \pm 0.06$ |
| | $21 \to 22$ | -11.55M | $86.55 \pm 0.01$ | $86.72 \pm 0.24$ | $85.20 \pm 0.26$ | $86.82 \pm 0.31$ |
| | original | 304.35M | $86.32 \pm 0.08$ | $86.32 \pm 0.08$ | $86.32 \pm 0.08$ | $86.32 \pm 0.08$ |
| ImageNet1k | $3 \to 4, 20 \to 23$ | -46.19M | $73.49 \pm 0.18$ | $74.03 \pm 0.43$ | $69.49 \pm 0.24$ | $73.68 \pm 0.12$ |
| | $17 \to 23$ | -69.28M | $84.58 \pm 0.19$ | $86.20 \pm 0.11$ | $84.83 \pm 0.31$ | $86.49 \pm 0.08$ |
| | $3 \to 4, 19 \to 23$ | -57.74M | $70.97 \pm 0.42$ | $71.50 \pm 0.14$ | $66.19 \pm 0.17$ | $70.75 \pm 0.07$ |
| | $20 \to 23$ | -34.64M | $74.45 \pm 0.07$ | $74.45 \pm 0.24$ | $70.19 \pm 0.30$ | $74.46 \pm 0.22$ |
| | $3 \to 4, 21 \to 23$ | -34.64M | $75.80 \pm 0.26$ | $76.30 \pm 0.14$ | $73.23 \pm 0.29$ | $76.14 \pm 0.22$ |
| | $20 \to 22$ | -23.09M | $75.49 \pm 0.25$ | $74.84 \pm 0.21$ | $70.56 \pm 0.25$ | $75.59 \pm 0.18$ |
| | $3 \to 4, 21 \to 22$ | -23.09M | $76.25 \pm 0.02$ | $76.61 \pm 0.29$ | $73.52 \pm 0.40$ | $76.43 \pm 0.21$ |
| | $20 \to 21$ | -11.55M | $77.00 \pm 0.27$ | $77.19 \pm 0.25$ | $72.72 \pm 0.31$ | $76.24 \pm 0.21$ |
| | $21 \to 22$ | -11.55M | $77.24 \pm 0.28$ | $77.06 \pm 0.24$ | $74.20 \pm 0.48$ | $77.14 \pm 0.27$ |
| | original | 304.35M | $78.55 \pm 0.20$ | $86.32 \pm 0.20$ | $86.32 \pm 0.20$ | $86.32 \pm 0.20$ |

> **Takeaway** TOAST consistently matches or outperforms deeper trained approximators while requiring no gradient-based training.

### 4.3 TOAST Applicability to other Tasks or Domains

We further evaluate TOAST beyond vision classification by applying it to text classification and semantic segmentation tasks. For text classification, we use `ModernBERT-B` on the `AG News` dataset, while for segmentation we employ the same backbone on the `SceneParse150` dataset.

Table 5: **TOAST Text Classification Performance on `AG News`**. Text classification accuracy, GFLOPs, and throughput for `ModernBERT-B` using `AG News`. The "Approx." column specifies the block mapping (output of the first block is used to approximate the output of the second). MLP is a trained approximators, while Linear is closed-form and training-free. Results are averaged over three seeds. Additional results are provided in Section A.2.4.

| Approx. | Params ↓ | Linear | | | MLP | | |
| | | Accuracy% ↑ | GFLOPs ↓ | img/s ↑ | Accuracy% ↑ | GFLOPs ↓ | token/s ↑ |
|---|---|---|---|---|---|---|---|
| $11 \to 21$ | 92.82M | $0.81 \pm 0.05$ | 12.7 | 2264.0 | $0.73 \pm 0.00$ | 12.68 | 2216.50 |
| $4 \to 8, 11 \to 14, 18 \to 21$ | 92.82M | $0.82 \pm 0.07$ | 12.7 | 2220.7 | $0.73 \pm 0.01$ | 12.68 | 2155.16 |
| $4 \to 7, 18 \to 21$ | 109.68M | $0.82 \pm 0.07$ | 15.9 | 1803.9 | $0.71 \pm 0.02$ | 15.85 | 1771.80 |
| $4 \to 8$ | 126.54M | $0.86 \pm 0.02$ | 19.0 | 1636.0 | $0.82 \pm 0.01$ | 19.03 | 1632.65 |
| $11 \to 14$ | 132.16M | $0.86 \pm 0.02$ | 20.1 | 1544.3 | $0.82 \pm 0.01$ | 20.08 | 1540.23 |
| $18 \to 21$ | 132.16M | $0.85 \pm 0.02$ | 20.1 | 1472.8 | $0.82 \pm 0.01$ | 20.08 | 1467.56 |
| $4 \to 5$ | 143.40M | $0.88 \pm 0.00$ | 22.2 | 1380.3 | $0.81 \pm 0.01$ | 22.20 | 1384.42 |
| $11 \to 12$ | 143.40M | $0.87 \pm 0.02$ | 22.2 | 1378.8 | $0.82 \pm 0.01$ | 22.20 | 1394.63 |
| $20 \to 21$ | 143.40M | $0.87 \pm 0.02$ | 22.2 | 1340.2 | $0.84 \pm 0.00$ | 22.20 | 1332.27 |
| original | 149.01M | $0.88 \pm 0.00$ | 23.25 | 1337.25 | $0.88 \pm 0.00$ | 23.25 | 1347.46 |

Additional implementation details, including model and dataset specifications, computational resources, and software tools, are provided in Tables 7 and 8 and Sections A.1.5 to A.1.7, with complete results in Section A.2.4. For both domains, we adopt the same setup as in the vision experiments: block approximations are implemented via a shared linear map, identity, or small MLP transformation applied across all tokens, estimated using a subset of 500 training samples. In the text domain, a linear classifier is trained on top of the frozen backbone for 5 epochs over 3 seeds. For segmentation, a segmentation head is trained on the frozen backbone for 10 epochs over 3 seeds. The results in Table 5 show that, in this setting as well, the linear transformation outperforms the more complex MLP. Moreover, up to 10 blocks can be approximated (i.e., $11 \rightarrow 21$), substantially reducing GFLOPs, improving throughput, and decreasing model size, while incurring only a minimal drop in accuracy. Results in Table 6 further demonstrate that a linear transformation is sufficient even for a more complex task such as segmentation, indicating that appropriately selecting which layers to approximate enables model size reduction with minimal impact on downstream accuracy.

Table 6: **Segmentation Performance.** mIoU results for each single skip configuration using `ViT-S` and `DiNO-B`.

| | mIoU $\uparrow$ | | | |
| | ViT-S | | DiNO-B | |
| Approx. | Linear | MLP | Linear | MLP |
|---|---|---|---|---|
| $0 \rightarrow 1$ | 0.27 | 0.26 | 0.29 | 0.29 |
| $1 \rightarrow 2$ | 0.29 | 0.29 | 0.29 | 0.29 |
| $2 \rightarrow 3$ | 0.30 | 0.30 | 0.29 | 0.29 |
| $3 \rightarrow 4$ | 0.30 | 0.29 | 0.29 | 0.29 |
| $4 \rightarrow 5$ | 0.30 | 0.29 | 0.29 | 0.29 |
| $5 \rightarrow 6$ | 0.28 | 0.27 | 0.29 | 0.29 |
| $6 \rightarrow 7$ | 0.28 | 0.27 | 0.29 | 0.29 |
| $7 \rightarrow 8$ | 0.29 | 0.28 | 0.26 | 0.23 |
| $8 \rightarrow 9$ | 0.28 | 0.27 | 0.28 | 0.27 |
| $9 \rightarrow 10$ | 0.29 | 0.29 | 0.27 | 0.26 |
| $10 \rightarrow 11$ | 0.30 | 0.29 | 0.27 | 0.26 |
| original | 0.31 | | 0.29 | |

> **Takeaway** TOAST extends beyond vision and standard classification, demonstrating broader applicability across domains.

## 5 LIMITATIONS AND FUTURE WORK

While TOAST efficiently approximates transformer blocks, our current investigation has primarily focused on vision transformer architectures and their application to classification tasks with preliminary results also extending to segmentation and text classification. Future research will explore the applicability of TOAST to other modalities and to diverse downstream tasks (e.g., image reconstruction). Such an expansion will be crucial for testing the universality of the observed block-similarity phenomena and assessing TOAST's adaptability. Furthermore, we aim to expand the analysis of these block-level similarities. This involves investigating redundancies at finer granularities, such as within individual attention heads or feed-forward layers, and consistently and developing more principled and reliable metrics for automatically selecting which blocks to approximate. The heuristic used in the current work, while effective, is not yet fully accurate, and improving it could enable more consistent identification of approximation-friendly layers with minimal impact on downstream performance. Such advancements may lead to more refined, context-aware approximation strategies that further enhance model efficiency.

## 6 CONCLUSION

In this work, we first analyze the emergence of consistent block-wise representation similarities within pretrained foundation models and then propose a method to leverage these similarities to obtain smaller and more efficient yet performant models. To this end, we propose Transformer Optimization using Adaptive and Simple Transformations (TOAST), a novel method for easily approximate entire transformer blocks using a simple transformation, without requiring additional training or fine-tuning. Our extensive empirical evaluations across multiple pretrained vision models and datasets validate that TOAST significantly reduces model parameters while maintaining, and sometimes even improving, downstream task performance. Furthermore, TOAST's straightforward linear approach often achieves better results than existing strategies like block skipping, and can be more effective than complex, trained approximations. TOAST thus offers a practical and efficient method for streamlining foundation models, making them more computationally accessible, and towards deployment in resource-constrained scenarios such as on-device settings.

ETHICS STATEMENT

This work adheres to the ICLR Code of Ethics. Our study focuses exclusively on pre-trained vision models and publicly available datasets, with no human subjects or sensitive personal data involved. All experimental protocols comply with legal, privacy, and ethical standards for AI research. The methods proposed in this paper aim solely to improve computational efficiency, without introducing harm or enabling misuse.

REPRODUCIBILITY STATEMENT

To ensure reproducibility, we provide detailed descriptions of all experiments, model configurations, datasets, training procedures, and hyperparameters in the main text and Appendix (Sections A.1.5 to A.1.7 and A.2.2). Additionally, the full implementation of TOAST, including scripts for block approximation and evaluation, is included as anonymous supplementary material. All results reported in the paper can be reproduced using these resources.

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

# A APPENDIX

## A.1 IMPLEMENTATION DETAILS

This section details the experiments conducted in Section 4, providing information to reproduce them. Additionally, we provide the code as a zip file in the supplementary material.

### A.1.1 MODELS AND DATASETS

Table 7 contains the full list of the pretrained models, while Table 8 contains dataset information.

Table 7: **Pretrained models details.** Details of the pretrained feature extractors with their Hugging-Face key, their alias, and their latent space dimensionality.

| Modality | HuggingFace Model Name | Alias | Enc. Dim |
|---|---|---|---|
| Vision | WinKawaks/vit-tiny-patch16-224 | `ViT-T` (Dosovitskiy et al., 2021) | 192 |
| | WinKawaks/vit-small-patch16-224 | `ViT-S` (Dosovitskiy et al., 2021) | 384 |
| | facebook/dinov2-small | `DiNO-S` (Oquab et al., 2023) | 384 |
| | facebook/deit-small-patch16-224 | `DEiT-S` (Touvron et al., 2020) | 384 |
| | google/vit-base-patch16-224 | `ViT-B` (Dosovitskiy et al., 2021) | 768 |
| | facebook/dinov2-base | `DiNO-B` (Oquab et al., 2023) | 768 |
| | laion/CLIP-ViT-B-16-laion2B-s34B-b88K | `OpenCLIP-ViT-B` (Zhai et al., 2019) | 768 |
| | google/vit-large-patch16-224 | `ViT-L` (Dosovitskiy et al., 2021) | 1024 |
| Text | answerdotai/ModernBERT-base | `ModernBERT-B` (Warner et al., 2025) | 768 |

Table 8: **Dataset details.** Details of the HuggingFace datasets used in the classification and reconstruction experiments, with the associated number of classes.

| Modality | Name | Alias | # Classes |
|---|---|---|---|
| Vision | MNIST (Deng, 2012) | `MNIST` | 10 |
| | Fashion-MNIST (Xiao et al., 2017) | `F-MNIST` | 10 |
| | CIFAR-10 (Krizhevsky et al., 2009) | `CIFAR-10` | 10 |
| | CIFAR-100 (coarse) (Krizhevsky et al., 2009) | `CIFAR-100C` | 20 |
| | CIFAR-100 (fine) (Krizhevsky et al., 2009) | `CIFAR-100F` | 100 |
| | SceneParse150 (Zhou et al., 2017; 2016) | `SceneParse150` | 150 |
| | Imagenet-1k (Russakovsky et al., 2015) | `ImageNet1k` | 1000 |
| Text | AG News Zhang et al. (2015) | `AG News` | 4 |

### A.1.2 APPROXIMATORS

The first implementation, referred to as the Res-MLP, is composed of two normalization layers and a feedforward submodule. The first layer normalization processes the input tensor, followed by a feedforward submodule comprising a linear transformation, a SiLU activation, a dropout layer, and a final linear transformation. The output of the feedforward submodule is added to the normalized input via a residual connection. This sum is then passed through the second normalization layer to produce the final output. The second implementation, referred to as the MLP, is a simplified MLP that employs a sequential architecture with a first linear transformation that reduces the input dimensionality to half of the target dimension, followed by a GELU activation function for smooth non-linearity, and a final linear transformation that restores the reduced features to match the target dimensionality. Refer to Listings 1 and 2 for the code snippet of the two translators.

Listing 1: Python Code Snippet for the Res-MLP translator

```python
class ResMLP(nn.Module):
    def __init__(self, num_features: int, dropout_p: float):
```

```python
        super().__init__()

        self.norm1 = nn.LayerNorm(num_features)
        self.norm2 = nn.LayerNorm(num_features)

        self.ff = nn.Sequential(
            nn.Linear(num_features, num_features),
            nn.SiLU(),
            nn.Dropout(p=dropout_p),
            nn.Linear(num_features, num_features),
        )

    def forward(self, x: torch.Tensor) -> torch.Tensor:
        x_normalized = self.norm1(x)
        x_transformed = self.ff(x_normalized)
        return self.norm2(x_transformed + x_normalized)
```

Listing 2: Python Code Snippet for the MLP translator

```python
translation = nn.Sequential(
    nn.Linear(x.size(1), y.size(1) // 2),
    nn.GELU(),
    nn.Linear(y.size(1) // 2, y.size(1)),
)
```

### A.1.3 METRIC ABLATION

We introduce linear approximation error as a simple, stable, and sample-efficient criterion for identifying redundant transformer blocks, offering a practical alternative for guiding block approximation. This metric measures how well the representation of a later block can be reconstructed from an earlier one through a least-squares projection, providing a direct estimate of how much additional structure the skipped layers contribute. Importantly, the error can be estimated using as few as $50$ samples producing substantially more stable and interpretable rankings compared to other metrics.

Table 9: **Top-5 Block Approximation Recommendation**. Top 5 recommended blocks to be approximated based on linear approximation error using `DEiT-S` and `CIFAR-100F`.

| Rank | Approx | # Layers | Predicted Error | Accuracy % |
|------|--------|----------|-----------------|------------|
| 1 | $9 \rightarrow 10$ | 1 | 0.14 | $71.69 \pm 0.11$ |
| 2 | $10 \rightarrow 11$ | 1 | 0.18 | $71.17 \pm 0.19$ |
| 3 | $8 \rightarrow 9$ | 1 | 0.23 | $70.83 \pm 0.13$ |
| 4 | $9 \rightarrow 11$ | 2 | 0.25 | $71.14 \pm 0.15$ |
| 5 | $8 \rightarrow 10$ | 2 | 0.26 | $71.06 \pm 0.19$ |
| - | original | 0 | - | 71.1 |

As shown in Table 9, linear approximation error correlates strongly with the actual accuracy impact of skipping or approximating a block range: blocks with the lowest error consistently incur minimal or no downstream performance degradation. This makes the metric both computationally lightweight and practically reliable for identifying redundant or compressible transformer regions.

To further validate this choice, we conduct an ablation comparing several candidate similarity metrics (e.g., cosine distance, MSE, Euclidean distance, and CKA) and evaluate how well each predicts the true accuracy drop after approximation. Results, summarized in Table 10, show that linear approximation error achieves the most consistent performance across architectures, with competitive or superior Precision@5 and Recall@5 scores. Notably, metrics such as cosine distance and Euclidean distance exhibit behavior that is highly model-dependent, while CKA performs well in some cases but is less stable across architectures and budgets.

Overall, this ablation highlights that linear approximation error provides the best trade-off between stability, computational cost, and predictive fidelity, making it a strong default metric for block selection in transformer approximation.

Table 10: **Block Selection Strategy Ablation.** Ranking evaluation metrics for approximation quality prediction on CIFAR-100 using `DEiT-S`, `DiNO-S`, and `DiNO-B`. Precision@5 and Recall@5 are shown for each model.

| | DEiT-S | | DiNO-S | | DiNO-B | | Mean | |
|---|---|---|---|---|---|---|---|---|
| | P@5 | R@5 | P@5 | R@5 | P@5 | R@5 | P@5 | R@5 |
| Linear Error | 0.6 | 0.6 | 0.6 | 0.6 | 0.6 | 0.6 | 0.60 | 0.60 |
| Cosine | 1.0 | 1.0 | 0.4 | 0.4 | 0.2 | 0.2 | 0.53 | 0.53 |
| CKA | 0.6 | 0.6 | 0.6 | 0.6 | 0.4 | 0.4 | 0.53 | 0.53 |
| MSE | 0.0 | 0.0 | 0.4 | 0.4 | 0.6 | 0.6 | 0.33 | 0.33 |
| Euclidean | 0.8 | 0.8 | 0.4 | 0.4 | 0.4 | 0.4 | 0.53 | 0.53 |

### A.1.4 BLOCK SELECTION PSEUDOCODE

---

**Algorithm 1** Identify Top-$k$ Layer Skip Configurations

---

**Require:** Model encoder $\mathcal{M}$ with $L$ layers, dataset $\mathcal{D}$, number of top configurations $k$, skip budget $b$ (optional)
**Ensure:** Top-$k$ skip configurations $\mathcal{S} = \{(s_1, e_1), \ldots, (s_k, e_k)\}$
 1: Extract layer representations: $\mathbf{H}_i \leftarrow \text{encode}(\mathcal{M}, \mathcal{D}, \text{layer}_i)$ for $i \in [0, L]$
 2: Initialize error matrix $\mathbf{E} \in \mathbb{R}^{L \times L}$
 3: **for** $i = 0$ to $L - 1$ **do**
 4:     **for** $j = i + 1$ to $L$ **do**
 5:         $\mathbf{E}_{i,j} \leftarrow \text{LinearApproximationError}(\mathbf{H}_i, \mathbf{H}_j)$
 6:     **end for**
 7: **end for**
 8: Initialize candidate list $\mathcal{C} \leftarrow \emptyset$
 9: **for** $i = 0$ to $L - 1$ **do**
10:     **for** $j = i + 1$ to $L$ **do**
11:         **if** $b$ is specified and $j - i \neq b$ **then**
12:             **continue**         ▷ Skip if not matching budget
13:         **end if**
14:         $\mathcal{C} \leftarrow \mathcal{C} \cup \{(i, j, \mathbf{E}_{i,j})\}$
15:     **end for**
16: **end for**
17: Sort $\mathcal{C}$ by error in ascending order
18: $\mathcal{S} \leftarrow$ top-$k$ configurations from $\mathcal{C}$
19: **return** $\mathcal{S}$

---

**Algorithm 2** Linear Approximation Error

---

**Require:** Source layer representations $\mathbf{X} \in \mathbb{R}^{n \times d}$, target layer representations $\mathbf{Y} \in \mathbb{R}^{n \times d}$
**Ensure:** Normalized residual error $\epsilon$
 1: Solve least-squares: $\mathbf{W}^* = \arg\min_{\mathbf{W}} \|\mathbf{Y} - \mathbf{X}\mathbf{W}\|_F^2$
 2: Compute prediction: $\hat{\mathbf{Y}} = \mathbf{X}\mathbf{W}^*$
 3: Compute normalized error: $\epsilon = \frac{\|\mathbf{Y} - \hat{\mathbf{Y}}\|_F}{\|\mathbf{Y}\|_F}$
 4: **return** $\epsilon$

---

### A.1.5 TOOLS & TECHNOLOGIES

All the experiments presented in this work employ the following tools:

- *PyTorch Lightning*, to ensure reproducible results while also getting a clean and modular codebase;

- *NN-Template GrokAI (2021)*, to easily bootstrap the project and enforce best practices;

Table 11: **Top-3 Block Approximation Recommendation**. Top 3 recommended blocks to be approximated based on linear approximation error and number of blocks to skip using `DEiT-S` and `CIFAR-100F`.

| # Blocks | Rank | Approx. | Predicted Error | Accuracy % |
|---|---|---|---|---|
|   | 1 | $9 \to 10$ | 0.14 | $71.69 \pm 0.11$ |
| 1 | 2 | $10 \to 11$ | 0.18 | $71.17 \pm 0.19$ |
|   | 3 | $8 \to 9$ | 0.23 | $70.83 \pm 0.13$ |
|   | 1 | $9 \to 11$ | 0.25 | $71.14 \pm 0.15$ |
| 2 | 2 | $8 \to 10$ | 0.26 | $71.06 \pm 0.19$ |
|   | 3 | $7 \to 9$ | 0.36 | $69.00 \pm 0.43$ |
|   | 1 | $8 \to 11$ | 0.36 | $68.22 \pm 0.40$ |
| 3 | 2 | $7 \to 10$ | 0.38 | $69.08 \pm 0.24$ |
|   | 3 | $6 \to 9$ | 0.45 | $65.64 \pm 0.03$ |
| 0 | - | original | - | 71.1 |

- *Transformers by HuggingFace*, to get ready-to-use transformers for both text and images;
- *Datasets by HuggingFace*, to access most of the datasets;
- *DVC* (Kuprieiev et al., 2022), for data versioning;
- *fvcore analysis library* (), for calculating GFLOPs;

### A.1.6 COMPUTATIONAL RESOURCES

Experiments involving larger models, specifically `DiNO-B`, `OpenCLIP-ViT-B`, and `ViT-L`, were conducted on an NVIDIA H100 GPU equipped with 93 GB of memory. All the other experiments utilized an NVIDIA GeForce RTX 5090 GPU with 31 GB of memory.

### A.1.7 EFFICIENCY METRICS

We evaluated model efficiency using two primary metrics. GFLOPs were used to measure the hardware-independent theoretical complexity of a single forward pass, calculated using the `fvcore` analysis library. Throughput, measured in samples per second, was used to quantify the practical, hardware-dependent inference speed. This was benchmarked by averaging the wall-clock time over numerous iterations on a single NVIDIA H100 GPU with a consistent batch size of 256.

### A.2 ADDITIONAL EXPERIMENTS

This section presents supplementary experiments to extend those detailed in Section 4.

### A.2.1 LATENT ANALYSIS

This section extend the analysis conducted in Section 4.1, to analyze block-wise internal similarities, to additional models of different dimensionality: `ViT-T`, `ViT-S`, `ViT-B` and `DiNO-S`. Additionally, we provide visualization using PCA for `DiNO-S`, `DEiT-S`, `ViT-S`, with different datasets and approximating both early and late blocks (see Figures 7 to 11).

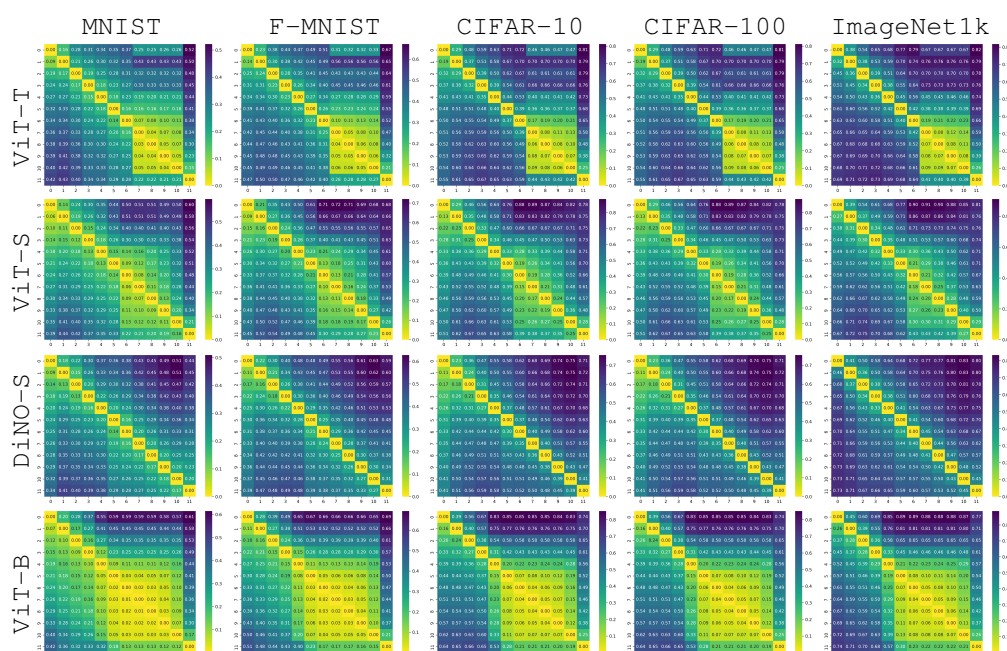

Figure 6: **Block Similarities:** Block-by-block similarities in `ViT-T`, `ViT-S`, `DiNO-S` and `ViT-B` models across five datasets: `MNIST`, `F-MNIST`, `CIFAR-10`, `CIFAR-100` and `ImageNet1k`. Each matrix quantifies the linear error between latent representations of different blocks, showing potential blocks for approximation. The matrices reveal that the similarity between blocks is predominantly influenced by the model rather than the specific dataset.

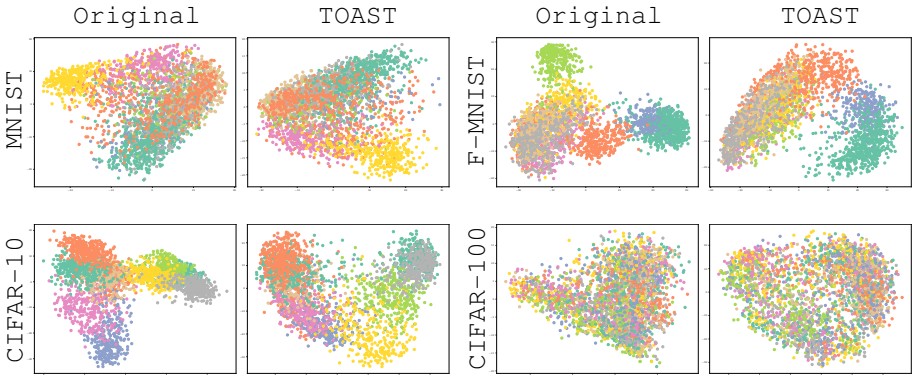

Figure 7: **Last Block Approximation.** PCA visualization of the final layer representations for both the original model and the model with its last block approximated from the preceding one. The representations are generated using the `DiNO-S` model across four datasets. The plots highlight that the last layer representations in this model are crucial, making it more effective to approximate earlier blocks instead. Note that for `CIFAR-100` (bottom right), only the overall structure of the space can be observed, as the 100 classes make it challenging to distinguish labels based on color.

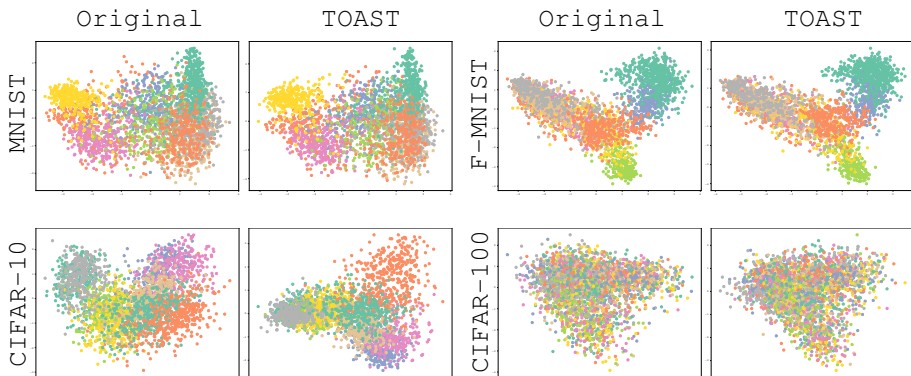

Figure 8: **Last Block Approximation.** PCA visualization of the final layer representations for both the original model and the model with its last block approximated by the preceding one. The representations are generated using the `DEiT-S` model across four datasets. The plots highlight that in this model, the representations in the last layer are redundant and can be effectively approximated, offering potential performance improvements while reducing model complexity and parameter count. Note that for `CIFAR-100` (bottom right), only the overall structure of the space can be observed, as the 100 classes make it challenging to distinguish labels based on color.

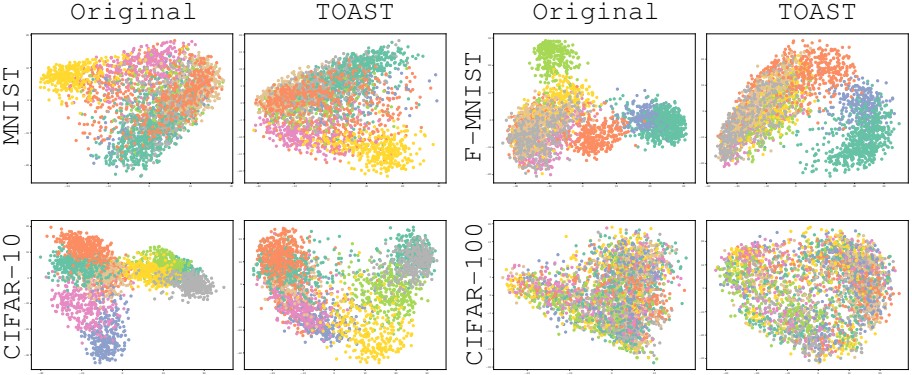

Figure 9: **Last Block Approximation.** PCA visualization of the final layer representations for both the original model and the model with its second block approximated by the preceding one. The representations are generated using the `DiNO-S` model across four datasets. Note that for `CIFAR-100` (bottom right), only the overall structure of the space can be observed, as the 100 classes make it challenging to distinguish labels based on color.

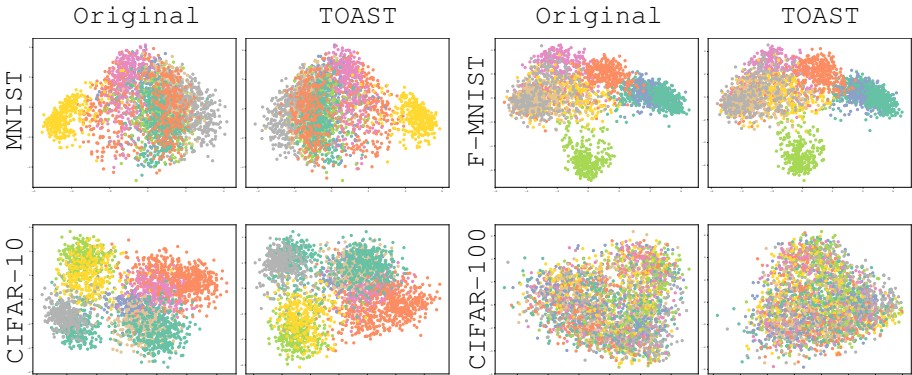

Figure 10: **Last Block Approximation.** PCA visualization of the last layer representations for both the original model and the model with its second block approximated using the previous one. Representations refer to the using `ViT-S` model across four datasets.

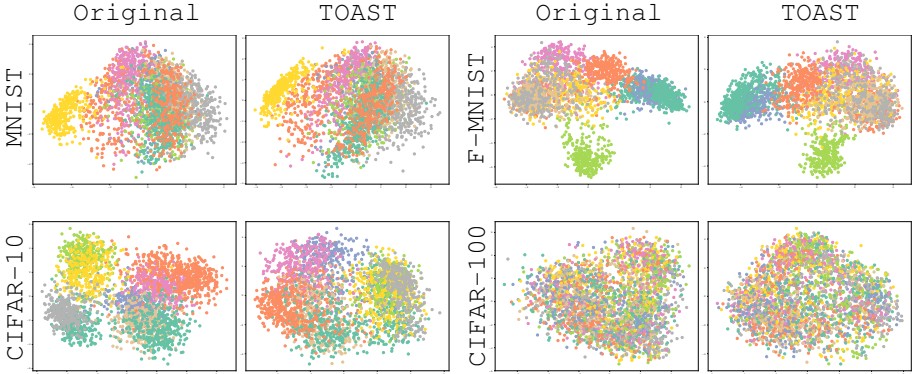

Figure 11: **Last Block Approximation.** PCA visualization of the last layer representations for both the original model and the model with its last block approximated from the previous one. Representations refer to the using `ViT-S` model across four datasets.

### A.2.2 IMAGE CLASSIFICATION

This section presents additional experiments that complement and extend those detailed in Section 4.2. Datasets and models are the ones detailed in Tables 7 and 8.

Table 12: **ViT-S Image Classification Performance Across Seeds.** Classification accuracy scores for ViT-S using multiple datasets, and 3 seeds. The "Approx." column specifies the blocks used for approximation, where the first value represents the block whose output is used to approximate the second block's output, while the "Params." column shows the number of parameters removed by the approximation compared to the original model.

| Approx. | Params. | MNIST | F-MNIST | CIFAR-10 | CIFAR-100C | CIFAR-100F | ImageNet1k |
|---------|---------|-------|---------|----------|-----------|-----------|-----------|
| $1 \to 5$ | 15.31M | $92.28 \pm 0.81$ | $86.90 \pm 0.72$ | $85.07 \pm 0.55$ | $68.01 \pm 0.31$ | $59.21 \pm 0.12$ | $44.04 \pm 0.42$ |
| $2 \to 5$ | 16.94M | $94.76 \pm 0.20$ | $88.57 \pm 0.31$ | $91.01 \pm 0.37$ | $77.77 \pm 0.22$ | $69.75 \pm 0.36$ | $60.38 \pm 0.12$ |
| $7 \to 10$ | 16.94M | $94.58 \pm 0.28$ | $88.44 \pm 0.35$ | $87.36 \pm 0.17$ | $72.58 \pm 0.69$ | $62.03 \pm 0.56$ | $35.80 \pm 0.11$ |
| $1 \to 3$ | 18.56M | $94.60 \pm 0.78$ | $88.36 \pm 0.44$ | $91.97 \pm 0.16$ | $79.36 \pm 0.54$ | $72.41 \pm 0.08$ | $64.99 \pm 0.29$ |
| $2 \to 4$ | 18.56M | $95.08 \pm 0.18$ | $88.83 \pm 0.21$ | $92.86 \pm 0.11$ | $81.45 \pm 0.44$ | $74.43 \pm 0.27$ | $67.52 \pm 0.16$ |
| $3 \to 5$ | 18.56M | $94.75 \pm 0.57$ | $88.81 \pm 0.19$ | $94.09 \pm 0.06$ | $83.16 \pm 0.34$ | $76.17 \pm 0.45$ | $67.27 \pm 0.45$ |
| $1 \to 2, 3 \to 4$ | 18.56M | $94.68 \pm 0.69$ | $88.30 \pm 0.25$ | $91.91 \pm 0.25$ | $79.72 \pm 0.16$ | $72.17 \pm 0.15$ | $65.38 \pm 0.03$ |
| $1 \to 2, 4 \to 5$ | 18.56M | $94.58 \pm 0.77$ | $88.95 \pm 0.07$ | $92.29 \pm 0.28$ | $80.14 \pm 0.10$ | $72.45 \pm 0.35$ | $64.42 \pm 0.24$ |
| $0 \to 1$ | 20.43M | $\mathbf{95.69} \pm 0.29$ | $88.81 \pm 0.19$ | $93.68 \pm 0.22$ | $83.55 \pm 0.23$ | $76.49 \pm 0.29$ | $65.11 \pm 0.27$ |
| $1 \to 2$ | 20.43M | $95.40 \pm 0.57$ | $88.53 \pm 0.62$ | $93.90 \pm 0.11$ | $83.98 \pm 0.22$ | $76.99 \pm 0.26$ | $70.32 \pm 0.38$ |
| $2 \to 3$ | 20.43M | $95.43 \pm 0.45$ | $88.93 \pm 0.62$ | $94.90 \pm 0.26$ | $85.72 \pm 0.48$ | $78.96 \pm 0.05$ | $71.26 \pm 0.03$ |
| $3 \to 4$ | 20.43M | $95.43 \pm 0.39$ | $88.77 \pm 0.36$ | $95.05 \pm 0.17$ | $85.99 \pm 0.37$ | $79.49 \pm 0.32$ | $\mathbf{71.40} \pm 0.22$ |
| $4 \to 5$ | 20.43M | $95.39 \pm 0.35$ | $89.18 \pm 0.51$ | $\mathbf{95.41} \pm 0.12$ | $86.27 \pm 0.27$ | $\mathbf{79.61} \pm 0.14$ | $70.98 \pm 0.16$ |
| $5 \to 6$ | 20.43M | $95.14 \pm 0.56$ | $89.30 \pm 0.54$ | $94.89 \pm 0.27$ | $\mathbf{86.49} \pm 0.33$ | $79.29 \pm 0.19$ | $69.25 \pm 0.09$ |
| $6 \to 7$ | 20.43M | $95.11 \pm 0.42$ | $88.94 \pm 0.66$ | $94.81 \pm 0.26$ | $85.33 \pm 0.30$ | $78.06 \pm 0.17$ | $67.41 \pm 0.08$ |
| $7 \to 8$ | 20.43M | $95.64 \pm 0.46$ | $89.41 \pm 0.45$ | $94.50 \pm 0.34$ | $85.30 \pm 0.50$ | $78.03 \pm 0.12$ | $66.22 \pm 0.10$ |
| $8 \to 9$ | 20.43M | $95.36 \pm 0.47$ | $\mathbf{89.64} \pm 0.37$ | $94.36 \pm 0.14$ | $84.66 \pm 0.25$ | $77.88 \pm 0.20$ | $64.03 \pm 0.29$ |
| $9 \to 10$ | 20.43M | $95.52 \pm 0.41$ | $89.57 \pm 0.10$ | $94.58 \pm 0.27$ | $81.76 \pm 0.34$ | $76.45 \pm 0.22$ | $61.82 \pm 0.24$ |
| $10 \to 11$ | 20.43M | $94.83 \pm 0.20$ | $89.11 \pm 0.43$ | $94.08 \pm 0.27$ | $82.13 \pm 0.70$ | $77.45 \pm 0.29$ | $63.92 \pm 0.25$ |
| original | 22.06M | $\underline{95.59} \pm 0.42$ | $\underline{89.04} \pm 0.85$ | $\underline{95.68} \pm 0.24$ | $\underline{87.61} \pm 0.39$ | $\underline{81.50} \pm 0.39$ | $\underline{73.24} \pm 0.13$ |

Table 13: **DiNO-S Image Classification Performance Across Seeds.** Classification accuracy scores for DiNO-S using multiple datasets, and 3 seeds. The "Approx." column specifies the blocks used for approximation, where the first value represents the block whose output is used to approximate the second block's output, while the "Params." column shows the number of parameters removed by the approximation compared to the original model.

| Approx. | Params. | MNIST | F-MNIST | CIFAR-10 | CIFAR-100C | CIFAR-100F | ImageNet1k |
|---------|---------|-------|---------|----------|-----------|-----------|-----------|
| $1 \to 5$ | 15.31M | $96.25 \pm 0.30$ | $86.50 \pm 1.42$ | $80.11 \pm 0.95$ | $59.15 \pm 0.45$ | $51.24 \pm 0.51$ | $18.70 \pm 0.09$ |
| $2 \to 5$ | 16.94M | $95.86 \pm 0.52$ | $87.99 \pm 0.30$ | $85.28 \pm 0.99$ | $67.50 \pm 1.02$ | $59.57 \pm 0.45$ | $40.63 \pm 0.59$ |
| $7 \to 10$ | 16.94M | $96.05 \pm 1.44$ | $88.28 \pm 1.25$ | $91.00 \pm 0.82$ | $78.47 \pm 0.61$ | $70.56 \pm 0.25$ | $45.66 \pm 0.69$ |
| $1 \to 3$ | 18.56M | $96.61 \pm 0.34$ | $88.48 \pm 0.61$ | $91.73 \pm 0.36$ | $78.62 \pm 0.87$ | $72.33 \pm 0.37$ | $56.85 \pm 0.21$ |
| $2 \to 4$ | 18.56M | $96.79 \pm 0.58$ | $88.34 \pm 0.33$ | $91.31 \pm 0.16$ | $76.41 \pm 0.44$ | $69.71 \pm 0.31$ | $60.16 \pm 0.41$ |
| $3 \to 5$ | 18.56M | $96.76 \pm 1.02$ | $88.65 \pm 0.92$ | $91.00 \pm 0.49$ | $75.51 \pm 0.45$ | $69.31 \pm 0.05$ | $57.47 \pm 0.11$ |
| $1 \to 2, 3 \to 4$ | 18.56M | $96.71 \pm 0.62$ | $88.69 \pm 0.46$ | $92.57 \pm 0.54$ | $79.16 \pm 1.02$ | $72.88 \pm 0.57$ | $59.79 \pm 0.19$ |
| $1 \to 2, 4 \to 5$ | 18.56M | $96.81 \pm 0.31$ | $88.67 \pm 1.23$ | $93.50 \pm 0.26$ | $79.35 \pm 1.00$ | $73.55 \pm 0.38$ | $58.62 \pm 0.25$ |
| $0 \to 1$ | 20.43M | $96.71 \pm 0.79$ | $88.97 \pm 1.12$ | $\mathbf{95.67} \pm 0.12$ | $\mathbf{85.89} \pm 0.56$ | $\mathbf{80.15} \pm 0.35$ | $61.25 \pm 0.22$ |
| $1 \to 2$ | 20.43M | $96.69 \pm 0.90$ | $88.26 \pm 1.10$ | $95.38 \pm 0.09$ | $84.86 \pm 0.84$ | $79.38 \pm 0.23$ | $64.86 \pm 0.36$ |
| $2 \to 3$ | 20.43M | $96.42 \pm 0.36$ | $88.31 \pm 1.20$ | $94.71 \pm 0.33$ | $84.15 \pm 0.94$ | $77.74 \pm 0.85$ | $65.16 \pm 0.69$ |
| $3 \to 4$ | 20.43M | $96.82 \pm 0.68$ | $88.77 \pm 0.78$ | $94.87 \pm 0.30$ | $83.96 \pm 0.62$ | $77.71 \pm 0.08$ | $\mathbf{65.35} \pm 0.56$ |
| $4 \to 5$ | 20.43M | $96.82 \pm 0.60$ | $89.15 \pm 0.72$ | $94.63 \pm 0.26$ | $83.04 \pm 0.62$ | $77.13 \pm 0.17$ | $64.28 \pm 0.24$ |
| $5 \to 6$ | 20.43M | $96.81 \pm 0.85$ | $88.75 \pm 0.86$ | $95.33 \pm 0.19$ | $84.83 \pm 0.04$ | $79.37 \pm 0.25$ | $64.88 \pm 0.43$ |
| $6 \to 7$ | 20.43M | $\mathbf{96.99} \pm 0.88$ | $\mathbf{89.42} \pm 0.68$ | $95.21 \pm 0.10$ | $83.82 \pm 0.53$ | $78.54 \pm 0.64$ | $63.61 \pm 0.62$ |
| $7 \to 8$ | 20.43M | $96.76 \pm 0.38$ | $89.05 \pm 1.24$ | $95.37 \pm 0.14$ | $84.57 \pm 0.42$ | $78.95 \pm 0.37$ | $61.59 \pm 0.31$ |
| $8 \to 9$ | 20.43M | $96.62 \pm 0.85$ | $88.45 \pm 1.21$ | $95.21 \pm 0.36$ | $84.98 \pm 0.22$ | $79.35 \pm 0.22$ | $61.73 \pm 0.43$ |
| $9 \to 10$ | 20.43M | $96.66 \pm 0.33$ | $88.53 \pm 0.71$ | $94.55 \pm 0.25$ | $83.97 \pm 1.25$ | $77.06 \pm 0.36$ | $58.56 \pm 0.25$ |
| $10 \to 11$ | 20.43M | $94.61 \pm 0.66$ | $86.96 \pm 1.18$ | $92.11 \pm 0.32$ | $79.85 \pm 0.26$ | $73.01 \pm 0.51$ | $50.76 \pm 0.33$ |
| original | 22.06M | $\underline{96.57} \pm 0.64$ | $\underline{88.07} \pm 1.40$ | $\underline{96.24} \pm 0.08$ | $\underline{87.53} \pm 0.45$ | $\underline{82.04} \pm 0.42$ | $\underline{67.45} \pm 0.45$ |

Table 14: **ViT-T Image Classification Performance.** Classification accuracy scores for `ViT-T` using multiple datasets, and 3 seeds. The "Approx." column specifies the blocks used for approximation, where the first value represents the block whose output is used to approximate the second block's output, while the "Params." column shows the number of parameters removed by the approximation compared to the original model.

| Approx. | Params. | MNIST | F-MNIST | CIFAR-10 | CIFAR-100C | CIFAR-100F | ImageNet1k |
|---|---|---|---|---|---|---|---|
| 1 →5 | 15.31M | 87.66 ± 0.57 | 85.10 ± 0.42 | 73.68 ± 0.46 | 53.46 ± 0.29 | 44.61 ± 0.42 | 22.21 ± 0.39 |
| 2 →5 | 16.94M | 90.59 ± 0.79 | 85.84 ± 0.18 | 82.41 ± 0.11 | 62.87 ± 0.21 | 54.68 ± 0.21 | 35.14 ± 0.38 |
| 7 →10 | 16.94M | 92.41 ± 0.47 | 86.50 ± 0.19 | 82.48 ± 0.85 | 69.26 ± 0.65 | 61.15 ± 0.28 | 39.03 ± 0.13 |
| 1 →3 | 18.56M | 90.55 ± 1.04 | 85.91 ± 0.22 | 80.48 ± 0.29 | 63.43 ± 0.25 | 54.57 ± 0.32 | 43.68 ± 0.26 |
| 2 →4 | 18.56M | 92.81 ± 0.56 | 86.58 ± 0.05 | 86.85 ± 0.17 | 70.49 ± 0.30 | 63.53 ± 0.23 | 49.94 ± 0.27 |
| 3 →5 | 18.56M | 91.84 ± 0.69 | 86.80 ± 0.04 | 88.00 ± 0.04 | 72.67 ± 0.30 | 65.66 ± 0.14 | 48.48 ± 0.37 |
| 1 →2, 3 →4 | 18.56M | 91.94 ± 0.78 | 86.71 ± 0.20 | 83.43 ± 0.41 | 66.92 ± 0.42 | 60.07 ± 0.48 | 45.14 ± 0.15 |
| 1 →2, 4 →5 | 18.56M | 90.86 ± 0.66 | 86.57 ± 0.24 | 84.61 ± 0.14 | 68.07 ± 0.55 | 60.11 ± 0.61 | 44.84 ± 0.26 |
| 0 →1 | 20.43M | 91.74 ± 0.48 | 86.22 ± 0.23 | 83.32 ± 0.22 | 68.58 ± 0.41 | 61.05 ± 0.36 | 44.12 ± 0.20 |
| 1 →2 | 20.43M | 91.65 ± 0.61 | 86.26 ± 0.24 | 85.84 ± 0.08 | 71.12 ± 0.06 | 63.85 ± 0.37 | 54.34 ± 0.44 |
| 2 →3 | 20.43M | 92.89 ± 0.18 | 86.49 ± 0.06 | 88.89 ± 0.08 | 74.90 ± 0.25 | 68.03 ± 0.37 | 57.83 ± 0.07 |
| 3 →4 | 20.43M | 93.10 ± 0.43 | **87.34** ± 0.03 | 89.73 ± 0.37 | 76.45 ± 0.17 | 70.04 ± 0.35 | 57.55 ± 0.14 |
| 4 →5 | 20.43M | 92.43 ± 0.20 | 87.22 ± 0.10 | 90.11 ± 0.32 | 76.40 ± 0.42 | 69.97 ± 0.37 | 55.91 ± 0.10 |
| 5 →6 | 20.43M | **93.57** ± 0.11 | 86.80 ± 0.13 | 90.17 ± 0.27 | 76.47 ± 0.35 | 70.69 ± 0.49 | 55.43 ± 0.38 |
| 6 →7 | 20.43M | 92.13 ± 0.37 | 86.77 ± 0.02 | 87.73 ± 0.22 | 72.35 ± 0.31 | 66.73 ± 0.45 | 47.39 ± 0.45 |
| 7 →8 | 20.43M | 93.20 ± 0.06 | 86.90 ± 0.30 | 88.58 ± 0.26 | 75.80 ± 0.29 | 69.28 ± 0.41 | 53.48 ± 0.24 |
| 8 →9 | 20.43M | 92.76 ± 0.11 | 87.18 ± 0.17 | 89.57 ± 0.33 | 76.43 ± 0.50 | 71.07 ± 0.33 | 56.07 ± 0.77 |
| 9 →10 | 20.43M | 92.39 ± 0.10 | 86.74 ± 0.18 | 89.86 ± 0.31 | 77.34 ± 0.04 | 71.70 ± 0.37 | 57.45 ± 0.29 |
| 10 →11 | 20.43M | 90.92 ± 0.48 | 86.89 ± 0.12 | **90.98** ± 0.21 | **78.85** ± 0.38 | **72.29** ± 0.42 | **58.94** ± 0.22 |
| original | 22.06M | 93.22 ± 0.18 | 86.99 ± 0.29 | 91.29 ± 0.06 | 79.27 ± 0.23 | 73.45 ± 0.38 | 63.02 ± 0.22 |

Table 15: **ViT-B Image Classification Performance.** Classification accuracy scores for `ViT-B` using multiple datasets, and 3 seeds. The "Approx." column specifies the blocks used for approximation, where the first value represents the block whose output is used to approximate the second block's output, while the "Params." column shows the number of parameters removed by the approximation compared to the original model.

| Approx. | Params. | Accuracy ↑ | | | | |
|---|---|---|---|---|---|---|
| | | MNIST | F-MNIST | CIFAR-10 | CIFAR-100C | CIFAR-100F |
| 1 → 5 | -25.99M | 87.06 ± 0.53 | 84.33 ± 0.61 | 73.54 ± 0.57 | 51.67 ± 1.10 | 38.98 ± 0.72 |
| 2 → 5 | -19.49M | 94.20 ± 0.21 | 87.80 ± 0.24 | 87.10 ± 0.83 | 71.68 ± 0.50 | 61.19 ± 0.37 |
| 1 → 3 | -13M | 96.51 ± 0.42 | 88.72 ± 0.41 | 93.71 ± 0.13 | 83.05 ± 0.23 | 74.74 ± 0.29 |
| 3 → 5 | -13M | 95.59 ± 0.09 | 88.28 ± 0.20 | 93.11 ± 0.06 | 83.50 ± 0.17 | 74.35 ± 0.47 |
| 2 → 4 | -13M | 96.21 ± 0.33 | 89.21 ± 0.64 | 94.59 ± 0.32 | 85.13 ± 0.24 | 76.82 ± 0.41 |
| 8 → 10 | -13M | 96.54 ± 0.21 | **89.72** ± 0.52 | 95.05 ± 0.26 | 85.78 ± 0.37 | 79.62 ± 0.14 |
| 9 → 11 | -13M | 95.59 ± 0.52 | 89.49 ± 0.26 | 93.22 ± 0.56 | 82.23 ± 0.44 | 76.33 ± 0.10 |
| 3 → 4 | -6.5M | 96.86 ± 0.35 | 89.69 ± 1.09 | **96.18** ± 0.09 | **89.18** ± 0.06 | **82.50** ± 0.17 |
| 4 → 5 | 6.5M | 96.55 ± 0.23 | 89.13 ± 0.50 | 95.39 ± 0.23 | 87.43 ± 0.15 | 80.30 ± 0.16 |
| 0 → 1 | -6.5M | 96.75 ± 0.29 | 88.97 ± 0.26 | 93.74 ± 0.15 | 84.49 ± 0.20 | 76.54 ± 0.29 |
| 1 → 2 | -6.5M | 96.88 ± 0.01 | 89.29 ± 0.24 | 95.63 ± 0.11 | 87.46 ± 0.20 | 80.64 ± 0.23 |
| 2 → 3 | -6.5M | **96.91** ± 0.17 | 89.69 ± 0.61 | 96.00 ± 0.18 | 88.38 ± 0.13 | 81.59 ± 0.35 |
| - | 86.39M | 95.61 ± 0.22 | 89.64 ± 0.57 | 96.25 ± 0.17 | 89.52 ± 0.23 | 83.41 ± 0.20 |

### A.2.3 ZERO-SHOT IMAGE CLASSIFICATION

To further assess the effectiveness of our approach, we evaluate TOAST in a zero-shot image classification setting. This evaluation utilizes the `OpenCLIP-ViT-B` model (Radford et al., 2021), which was pretrained on `LAION-2B` Schuhmann et al. (2022), with `ImageNet1k` serving as the downstream evaluation dataset. The analysis is conducted only on the base version, as larger versions (e.g., `OpenCLIP-ViT-L` or `OpenCLIP-ViT-H`) contain too many parameters and are thus beyond the scope of this paper. As in previous experiments, the model remains frozen, and block approximations are computed using a shared linear transformation applied across all tokens, based on a subset of 3,000 training samples. Importantly, we apply these approximations only to the vision encoder, leaving the text encoder unchanged. We follow the standard `ImageNet1k` prompt templates. The results in Table 16 lead to the conclusion that the impact on zero-shot accuracy is highly dependent on the targeted block's position. The choice of which blocks to approximate is therefore crucial. For instance, approximating an early block (e.g., $1 \rightarrow 2$ or $2 \rightarrow 3$) results in a modest accuracy drop (e.g., 5.51%), yielding a competitive model with fewer parameters. In contrast, approximating the final block (i.e., $10 \rightarrow 11$) causes a catastrophic performance collapse of 67.75%. Meaning that, for `OpenCLIP-ViT-B`, later layers in the vision encoder appear to capture uniquely critical information for zero-shot generalization that cannot be effectively replicated by earlier ones. To the best of our knowledge, this work is the first to investigate training-free model size reduction in this challenging setting.

Table 16: **Zero-shot image classification.** Accuracy scores for `OpenCLIP-ViT-B` on `ImageNet1k`. The "Approx." column specifies the blocks being approximated, where the first value represents the block whose output is used to approximate the second block's output. The "$\Delta$" column indicates the change in accuracy.

| Params. | Approx. | Accuracy ↑ | $\Delta$ |
|---------|---------|------------|----------|
| | $0 \rightarrow 1$ | 57.93 | -17.41% |
| | $1 \rightarrow 2$ | 64.20 | -8.56% |
| | $2 \rightarrow 3$ | **66.35** | **-5.51%** |
| | $3 \rightarrow 4$ | 64.65 | -7.90% |
| | $4 \rightarrow 5$ | 64.86 | -7.60% |
| -6.49M | $5 \rightarrow 6$ | 58.05 | -17.32% |
| | $6 \rightarrow 7$ | 61.56 | -12.31% |
| | $7 \rightarrow 8$ | 58.53 | -16.64% |
| | $8 \rightarrow 9$ | 52.32 | -25.50% |
| | $9 \rightarrow 10$ | 59.21 | -15.68% |
| | $10 \rightarrow 11$ | 22.64 | -67.75% |
| 149.07M | original | 70.21 | – |

### A.2.4 TOAST APPLICABILITY TO OTHER TASKS OR DOMAINS

This section presents additional experiments that complement and extend those detailed in Section 4.3. Datasets and models are the ones detailed in Tables 7 and 8.

Table 17: **TOAST Text Classification Performance on `AG News`**. Text classification accuracy, GFLOPs, and throughput for `ModernBERT-B` using `AG News`. The "Approx." column specifies the block mapping (output of the first block is used to approximate the output of the second). MLP is a trained approximators, while Linear is closed-form and training-free. Results are averaged over three seeds.

| Approx. | Params ↓ | Linear | | | MLP | | |
|---|---|---|---|---|---|---|---|
| | | Accuracy% ↑ | GFLOPs ↓ | img/s ↑ | Accuracy% ↑ | GFLOPs ↓ | img/s ↑ |
| $11 \to 21$ | 92.82M | $0.81 \pm 0.05$ | 12.7 | 2264.0 | $0.73 \pm 0.00$ | 12.68 | 2216.50 |
| $4 \to 8, 11 \to 14, 18 \to 21$ | 92.82M | $0.82 \pm 0.07$ | 12.7 | 2220.7 | $0.73 \pm 0.01$ | 12.68 | 2155.16 |
| $4 \to 7, 18 \to 21$ | 109.68M | $0.82 \pm 0.07$ | 15.9 | 1803.9 | $0.71 \pm 0.02$ | 15.85 | 1771.80 |
| $4 \to 8$ | 126.54M | $0.86 \pm 0.02$ | 19.0 | 1636.0 | $0.82 \pm 0.01$ | 19.03 | 1632.65 |
| $11 \to 14$ | 132.16M | $0.86 \pm 0.02$ | 20.1 | 1544.3 | $0.82 \pm 0.01$ | 20.08 | 1540.23 |
| $18 \to 21$ | 132.16M | $0.85 \pm 0.02$ | 20.1 | 1472.8 | $0.82 \pm 0.01$ | 20.08 | 1467.56 |
| $1 \to 2$ | 143.40M | $0.84 \pm 0.01$ | 22.2 | 1386.0 | $0.84 \pm 0.00$ | 22.20 | 1385.20 |
| $2 \to 3$ | 143.40M | $0.86 \pm 0.00$ | 22.2 | 1379.8 | $0.86 \pm 0.00$ | 22.20 | 1388.06 |
| $3 \to 4$ | 143.40M | $0.82 \pm 0.01$ | 22.2 | 1391.6 | $0.83 \pm 0.00$ | 22.20 | 1392.14 |
| $4 \to 5$ | 143.40M | $0.88 \pm 0.01$ | 22.2 | 1380.3 | $0.81 \pm 0.01$ | 22.20 | 1384.42 |
| $5 \to 6$ | 143.40M | $0.86 \pm 0.02$ | 22.2 | 1385.0 | $0.83 \pm 0.01$ | 22.20 | 1392.14 |
| $6 \to 7$ | 143.40M | $0.86 \pm 0.02$ | 22.2 | 1387.8 | $0.85 \pm 0.01$ | 22.20 | 1387.81 |
| $7 \to 8$ | 143.40M | $0.87 \pm 0.01$ | 22.2 | 1384.8 | $0.85 \pm 0.00$ | 22.20 | 1365.78 |
| $8 \to 9$ | 143.40M | $0.84 \pm 0.01$ | 22.2 | 1384.4 | $0.83 \pm 0.01$ | 22.20 | 1383.31 |
| $9 \to 10$ | 143.40M | $0.82 \pm 0.08$ | 22.2 | 1385.3 | $0.71 \pm 0.01$ | 22.20 | 1385.92 |
| $10 \to 11$ | 143.40M | $0.81 \pm 0.08$ | 22.2 | 1383.2 | $0.72 \pm 0.03$ | 22.20 | 1381.78 |
| $11 \to 12$ | 143.40M | $0.87 \pm 0.02$ | 22.2 | 1378.8 | $0.82 \pm 0.01$ | 22.20 | 1394.63 |
| $12 \to 13$ | 143.40M | $0.86 \pm 0.02$ | 22.2 | 1384.5 | $0.83 \pm 0.01$ | 22.20 | 1390.65 |
| $13 \to 14$ | 143.40M | $0.80 \pm 0.06$ | 22.2 | 1385.2 | $0.73 \pm 0.02$ | 22.20 | 1385.23 |
| $14 \to 15$ | 143.40M | $0.84 \pm 0.04$ | 22.2 | 1390.0 | $0.79 \pm 0.01$ | 22.20 | 1387.43 |
| $15 \to 16$ | 143.40M | $0.85 \pm 0.02$ | 22.2 | 1402.7 | $0.82 \pm 0.00$ | 22.20 | 1381.80 |
| $16 \to 17$ | 143.40M | $0.87 \pm 0.01$ | 22.2 | 1402.8 | $0.85 \pm 0.00$ | 22.20 | 1387.02 |
| $17 \to 18$ | 143.40M | $0.85 \pm 0.02$ | 22.2 | 1402.3 | $0.83 \pm 0.01$ | 22.20 | 1389.71 |
| $18 \to 19$ | 143.40M | $0.87 \pm 0.01$ | 22.2 | 1403.5 | $0.85 \pm 0.01$ | 22.20 | 1393.53 |
| $19 \to 20$ | 143.40M | $0.85 \pm 0.02$ | 22.2 | 1403.9 | $0.82 \pm 0.00$ | 22.20 | 1390.19 |
| $20 \to 21$ | 143.40M | $0.87 \pm 0.02$ | 22.2 | 1340.2 | $0.84 \pm 0.00$ | 22.20 | 1332.27 |
| original | 149.01M | $0.88 \pm 0.00$ | 23.25 | 1337.25 | $0.88 \pm 0.00$ | 23.25 | 1347.46 |

### A.2.5 Evaluation with original classification heads

Table 18: **Comparison of Original vs. Retrained Classification Heads.** TOAST performance on `ImageNet1k` using the frozen, pre-trained head (Original) versus a linear classifier trained on the frozen backbone (Retrained). The relative ranking of approximations remains consistent across both settings.

| Encoder | Approximation | Original Head Acc. ↑ | Retrained Head Acc. ↑ |
|---|---|---|---|
| DEiT-S | $3 \to 4, 9 \to 11$ | 72.44 | $68.39 \pm 0.13$ |
| | $3 \to 4, 9 \to 10$ | 77.25 | $71.35 \pm 0.22$ |
| | $2 \to 3$ | 78.69 | $73.19 \pm 0.19$ |
| | $10 \to 11$ | 78.78 | $73.78 \pm 0.28$ |
| | original | 79.66 | $73.85 \pm 0.39$ |
| ViT-S | $1 \to 2$ | 76.62 | $70.32 \pm 0.38$ |
| | $2 \to 3$ | 78.25 | $71.26 \pm 0.03$ |
| | $3 \to 4$ | 78.25 | $71.40 \pm 0.22$ |
| | $4 \to 5$ | 77.66 | $70.98 \pm 0.16$ |
| | original | 79.86 | $73.24 \pm 0.13$ |

As mentioned in the main paper, our primary evaluation involves training a new linear classifier on top of the frozen model backbone to simulate a realistic transfer learning scenario. However, the original papers for `DEiT-S` (Touvron et al., 2021) and `ViT-S` (Beyer et al., 2022) report performance using the classification head that was part of the original pre-training.

To confirm that our conclusions are robust and not an artifact of our evaluation protocol, we conducted an additional set of experiments using the official, pre-trained classification heads from the original model checkpoints. For consistency with our main experiments, we use the same number of samples (500) for the approximation. In this setup, we do not train a new classifier; we simply evaluate the accuracy of the frozen, approximated models using their original heads.

The results, presented in Table 18, are fully consistent with the main conclusions of our paper. They confirm that our block approximation method provides a favorable accuracy-efficiency trade-off, even when evaluated with the original model heads. The relative drop in accuracy when approximating different layers follows the same patterns observed in our primary experiments, reinforcing the validity of our approach.

### A.2.6 Computational efficiency vs. accuracy

To quantify the effectiveness of different approximation methods, we analyze the trade-off between downstream accuracy and computational cost. Figure 12 presents this analysis on a `DiNO-B` model using both `CIFAR-100F` and `ImageNet1k` against three standard efficiency metrics: parameter count, GFLOPs, and inference throughput. Across all metrics, the proposed linear translator (green) establishes a more favorable Pareto frontier compared to the baseline identity-based approach (blue). This indicates that for any given efficiency budget (e.g., a specific GFLOPs target), the linear translator consistently yields a model with higher accuracy.

### A.2.7 Analysis of misclassifications

In this section, we examine changes in per-class accuracy and misclassification patterns. As shown in Figure 13, models behave differently at block approximations. `DiNO-S` remains remarkably stable across blocks and classes, with the only degradation appearing for classes dog (when approximating blocks 10 or 11) and deer (for block 10 approximation). `ViT-S` shows a similar drop for class dog on its final block. Instead, the most noticeable hit occurs for class cat when the earlier blocks are approximated. For `DEiT-S`, several mid-to-late block approximations improve accuracy for various classes, whereas the very first block causes a clear relative decline in nearly every class. These observations suggest strategies like preferring late-block approximation for `DEiT-S`, or reserving extra samples for the linear transformation in order to recover the accuracy of difficult classes for the model.

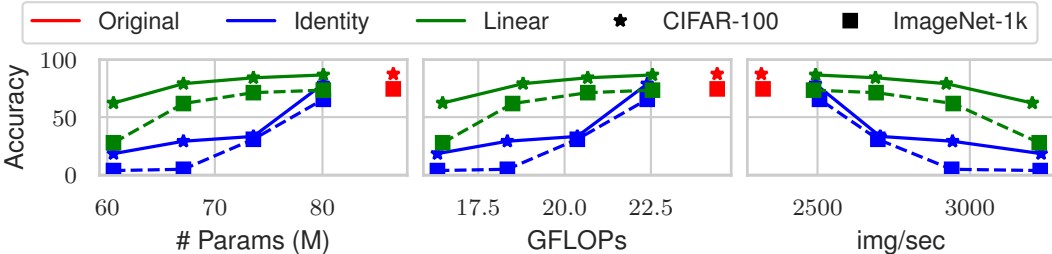

Figure 12: **Accuracy-efficiency trade-off for different approximation strategies.** Each subplot shows the accuracy against a different efficiency metric: the number of parameters (left), GFLOPs (center), and inference throughput (right). The image shows that the linear translator achieves a superior accuracy-efficiency trade-off.

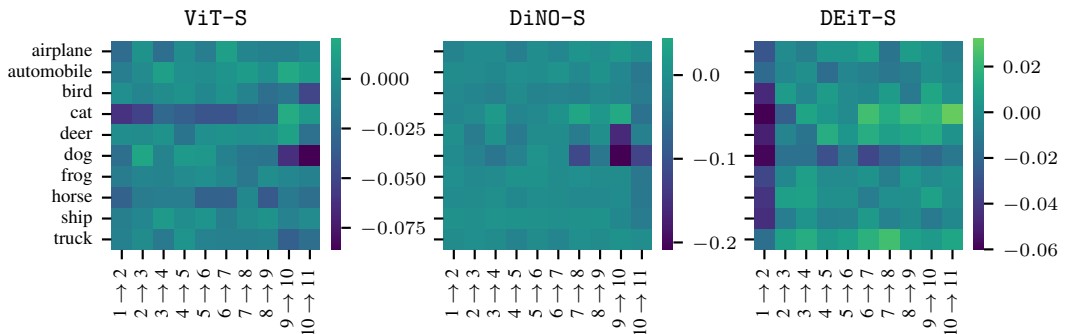

Figure 13: **Per-class accuracy delta on `CIFAR-10` when a single block is approximated in `ViT-S`, `DiNO-S` and `DEiT-S`**. Cell values indicate the relative change in the accuracy with respect to the original model. Brighter (green) cells indicate an accuracy gain for the class, while darker (blue) cells indicate an accuracy drop.

In order to further investigate how the predictions change while approximating blocks, we plot the difference in the normalized confusion matrix before and after the approximation. In Figure 14, we show the delta confusion matrix for `DEiT-S` on `CIFAR-100C`. Also, here we can see how approximating the very first block makes the model puzzling and lose per-class accuracy (i.e., negative delta along the diagonal). On the other hand, approximating the last block acts as a regularizer, resulting in an overall gain in the per-class accuracy and, as a consequence, fewer misclassifications (negative deltas off-diagonal). This supports results shown in Figure 13 and Table 3.

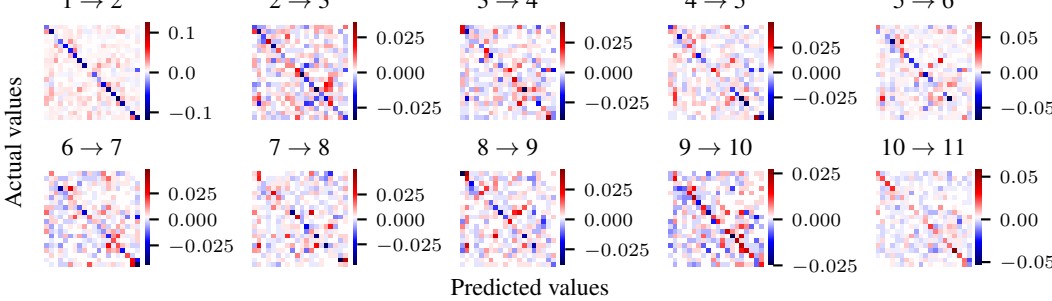

Figure 14: **Normalized relative confusion matrix when single blocks are approximated for `DEiT-S` on `CIFAR-100C`**. Diagonal cells capture the per-class change in accuracy, whereas off-diagonal cells capture changes in misclassifications between classes. Red (positive) values on the diagonal mean the approximation improves that class's accuracy. Red off-diagonal values mean more misclassifications. Conversely, blue (negative) off-diagonal values indicate fewer misclassifications, and blue values on the diagonal indicate a drop in per-class accuracy.

Additionally, Figure 15 shows representative CIFAR-10 images that become misclassified after approximating a block of ViT-S. The patterns we observe mirror the trends in Figures 13 and 14: when approximating earlier blocks, we observe many images belonging to class cat to be misclassified. Instead, when approximating later blocks, we observe images of the class dog to be misclassified. Together, these qualitative examples show that understanding these block-specific vulnerabilities allows us to steer the approximation procedure, informing choices about which blocks to approximate based on the observed impact on the final model's class-wise performance.

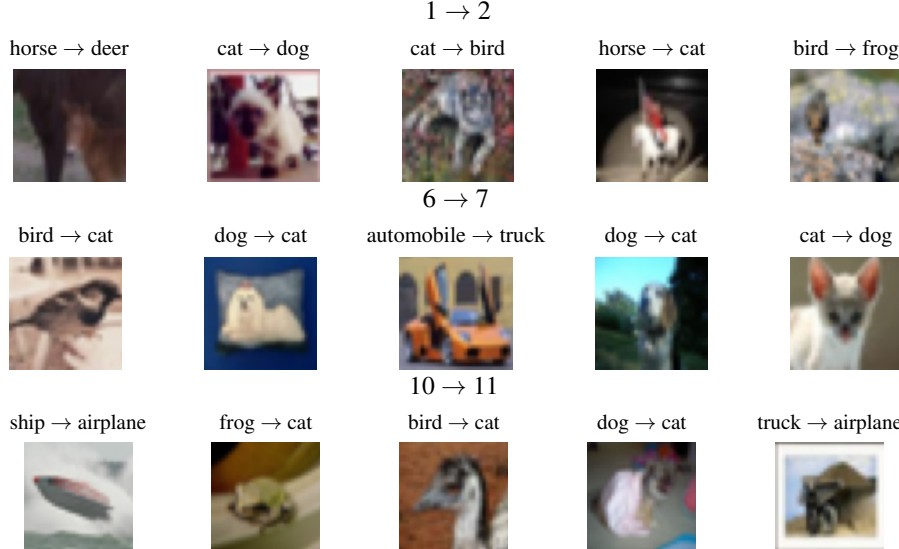

Figure 15: **Visualization of misclassified samples after approximating a block of ViT-S on CIFAR-10**. Images from CIFAR-10 whose label *flips from correct to incorrect* when specific blocks are approximated. The title reports the true class followed by the wrong prediction.

