# OpenReview forum: "TOAST : Transformer Optimization using Adaptive and Simple Transformations"
_ICLR.cc/2026/Conference — Submitted to ICLR 2026_

### Official Review · Reviewer_dwJk · 2025-10-15

**Soundness:** 2
**Presentation:** 3
**Contribution:** 1
**Rating:** 4
**Confidence:** 4

**Summary:**

This paper explores intra-network redundancy and introduces Transformer Optimization using Adaptive and Simple Transformations (TOAST), a framework that exploits these redundancies to approximate entire transformer blocks with lightweight, closed-form mappings, such as a linear transformation or even the identity function.

**Strengths:**

(1) The paper's approach to improving model efficiency from the perspective of intra-network redundancy is a promising and valuable area of research.

**Weaknesses:**

(1) Insufficient Comparison with Related Fields: The paper's connection to well-established fields like layer-wise pruning and Stitchable Neural Networks (SN-Nets) is not adequately discussed. Replacing a block with an identity or linear transformation is conceptually similar to these methods, yet the paper lacks a clear comparison, making it difficult to assess the relative advantages and novelty of TOAST.

(2) Lack of Methodological Clarity: The mechanism for selecting the appropriate approximation (identity or linear) for a given block is unclear. The paper does not provide a formal algorithm or a sufficiently detailed explanation of the decision process. This ambiguity hinders the reproducibility and a deeper understanding of the core mechanism.

(3) Limited Experimental Scope: The experimental evaluation is confined to the vision domain. While results on ViT are presented, there is no evidence to suggest that the proposed method is generalizable to other important domains, particularly Large Language Models (LLMs). This narrow scope limits the perceived impact and applicability of the work.

(4) Unconvincing Empirical Performance: The empirical evidence supporting the method's effectiveness is not compelling. As shown in Tables 1 and 2, the method often leads to significant performance degradation with only modest parameter reduction. The claimed benefits appear to be isolated to very large-scale models (ViT-L) or DEiT-S, and are not consistently demonstrated across different settings, which questions the practical utility of the approach.

(5) Inadequate Baseline Selection: The set of baselines used for comparison is insufficient. Given the conceptual overlap with model pruning, the paper should have included comparisons against relevant pruning methods, especially those from the vision literature. Without these comparisons, the claimed efficiency and performance benefits of TOAST are not well-contextualized.

**Questions:**

(1) The paper suggests replacing a full block with an identity transformation. Could this be interpreted as pruning the entire block? If so, how does this method differ from the well-established technical line of layer-wise pruning? Similarly, if a linear transformation is used, does this effectively replace the block with a single linear layer, thereby "stitching" blocks together? If so, what are the advantages of this approach compared to more efficient methods like Stitchable Neural Networks (SN-Nets)? Given that SN-Nets are known for their resource efficiency and flexibility, a more detailed discussion and comparison would be highly beneficial.

(2) Regarding the "Approximating Transformer Blocks" section, could you clarify the selection process between a linear transformation and an identity mapping? Is this choice optimized via a loss function, or is it determined by directly comparing which mapping results in a smaller loss? A formal algorithm or pseudocode describing this process would greatly improve clarity.

(3) The experiments are primarily conducted on vision backbones and tasks. Have the authors investigated whether the TOAST framework is equally effective for Large Language Model (LLM) backbones? such as Qwen and Llama.

(4) Based on Tables 1 and 2, the empirical results do not appear consistently strong. The method seems to maintain a slight performance advantage only on the very large ViT-L model and Deit-S while reducing parameters by about 10%. In most other cases, performance degrades substantially even with a parameter reduction of less than 10%. Does this suggest that the practical effectiveness of the method may be limited?

(5) In the baseline comparisons, would it be more appropriate to include baselines from the model pruning literature, particularly methods well-established in the vision domain? This seems especially relevant given the conceptual similarity between replacing a block with an identity mapping and block-level pruning.

---

> ### Author Response · Authors · 2025-11-21
> **[1/2]**
>
> We appreciate the reviewer’s assessment and their recognition that addressing intra-network redundancy is a promising direction for advancing model efficiency. We address their questions below:
>
> **W1, W5, Q1, Q5. Comparison with related fields:** We did not discuss SN-Nets in detail because we consider them to operate in a fundamentally different setting. SN-Nets build hybrid models by combining components from pre-trained models of different sizes (e.g., small, base, large), typically involving additional training. In contrast, TOAST operates within a single pre-trained network without any additional training. We did discuss related methods, such as pruning in the related work section, but we did not include direct comparisons because TOAST is specifically positioned within the post-hoc, training-free paradigm, targeting scenarios where users may lack the resources to fine-tune pre-trained models. Methods like pruning often require additional fine-tuning, which involves a substantially larger computational budget to recover performance. As a result, a direct performance comparison could be computationally unfair and potentially misleading. That said, we are happy to consider additional baselines or comparisons if the reviewer could suggest specific relevant works that would be appropriate within our post-hoc, training-free setting.
>
> **W2 & Q2. Methodological clarity:** In TOAST, we use, by default, a linear transformation to approximate a block. For some cases, we also explored the effect of just using an identity mapping, which is a special case of a linear mapping too. Surprisingly, we found this to work well too, particularly when the target block is highly redundant. However, the choice between the linear or the identity mapping is not part of TOAST itself, but rather an ablation study. Hence, it is not selected via any optimization or loss minimization, but rather an empirical observation that linear approximations generally perform well, while identity mappings can be used as a simpler alternative in specific cases.

---

> ### Author Response · Authors · 2025-11-21
> **[2/2]**
>
> **W3 & Q3. Additional experiments:**  We agree that evaluating TOAST beyond vision transformers is an interesting additional direction to assess its generality. Given time constraints, we conducted a preliminary experiment extending TOAST to text transformers, using ModernBERT-base[1] on the AG News[2] text-classification benchmark. We followed the same protocol as in our vision experiments: 500 training samples were used to fit a linear transformation between blocks with frozen backbone weights, and only a linear classifier was trained for 5 epochs using 3 different seeds. These results are not intended to achieve state-of-the-art performance, but rather to maintain methodological consistency and demonstrate the potential applicability of TOAST beyond the vision domain.
>
> The preliminary findings indicate that TOAST transfers effectively to non-vision domains and is architecture-agnostic, operating consistently across both vision and language models. For instance, approximating 10 blocks ($12 \rightarrow 22$) improves efficiency while slightly decreasing accuracy (87% vs. 83%). We will include additional results (including all the approximations) in the final manuscript, such as an ablation over different transformations (identity, linear, MLP), efficiency metrics (GFLOPS, throughput), and averages across multiple seeds.
>
> | Approximation       | Mean Accuracy
> |---------------------|------|
> | Baseline            | 0.87 |
> | $5 \rightarrow 9$   | 0.86 |
> | $12 \rightarrow 15$ | 0.86 |
> | $12 \rightarrow 22$ | 0.83 |
> | $18 \rightarrow 22$ | 0.84 |
> | $19 \rightarrow 22$ | 0.85 |
>
> Additionally, to further evaluate the proposed method, we conducted a preliminary experiment on semantic segmentation using the MIT Scene Parsing Benchmark[3], employing the same ViT-S backbone, as well as the same protocol as for the classification task. We used 500 samples to estimate the transformation and the pre-trained backbone is kept frozen, while only a simple segmentation head is trained. Performance are measured using the Mean Intersection over Union (mIoU) score. Given the time constraint, these results are not intended to reach SOTA performance, but rather to maintain methodological consistency and to demonstrate that TOAST is applicable beyond classification tasks. Although these experiments are not yet exhaustive, they follow the trends observed in our primary results and further support the generality of our approach.
> | Approximation       | Identity | Linear | MLP  |
> |---------------------|----------|--------|------|
> | Baseline            | 0.31     | 0.31   | 0.31 |
> | $0 \rightarrow 1$   | 0.18     | 0.27   | 0.27 |
> | $1 \rightarrow 2$   | 0.26     | 0.29   | 0.29 |
> | $2 \rightarrow 3$   | 0.23     | 0.30   | 0.30 |
> | $9 \rightarrow 10$  | 0.29     | 0.29   | 0.29 |
> | $10 \rightarrow 11$ | 0.30     | 0.30   | 0.30 |
> |---------------------|----------|--------|------|
> | $1 \rightarrow 3$   | 0.14    | 0.27   | 0.26
> | $2 \rightarrow 5$   | 0.11   | 0.25   | 0.24
> | $9 \rightarrow 11$   | 0.29    | 0.28   | 0.27
>
> Finally, we will update the limitations section to reflect that TOAST has now been preliminarily validated on text and segmentation tasks, while emphasizing that a comprehensive evaluation across multiple modalities and downstream tasks remains an important direction for future work.
>
> **W4 & Q4. Practical effectiveness:**  We would like to clarify a few points regarding the reported performance. As noted by the reviewer, performance improvements are observed in both ViT-L and DeiT-S. Since DeiT-S is much smaller than ViT-L, this suggests that the benefits are not limited to very large-scale models, and the method can be effective across a range of model sizes depending on the context. The slight performance degradation in some smaller models reflects the natural trade-off between parameter reduction and accuracy; even modest reductions (~10%) can impact low-capacity models. In general, our results show that TOAST consistently reduces parameters while minimally affecting performance, where efficiency gains are most valuable for practical deployment.
>
> We believe our responses address the reviewer’s concerns, and we are happy to provide further clarification if needed.
>
>
>
> ----
>
> [1] Warner, Benjamin, et al. Smarter, better, faster, longer: A modern bidirectional encoder for fast, memory efficient, and long context finetuning and inference.
>
> [2] Xiang Zhang et al., Character-level Convolutional Networks for Text Classification.
>
> [3] B. Zhou et al., Scene Parsing through ADE20K Dataset.

---

### Official Review · Reviewer_sH5D · 2025-10-27

**Soundness:** 2
**Presentation:** 3
**Contribution:** 2
**Rating:** 4
**Confidence:** 4

**Summary:**

Based on the observation that intermediate features of pretrained vision transformers are highly similar, the paper proposes Transformer Optimization using Adaptive and Simple Transformations (TOAST), which replace redundant blocks with lightweight transformations, e.g., identity operations or linear mappings. It uses a small number of training samples, e.g., 500, to find the optimal linear transformation that replaces the original transformer blocks. Experiments show that TOAST reduces parameters and computations while preserving the downstream performance.

**Strengths:**

1. The paper is well-organized and easy to follow. The proposed method is naturally based on existing observations. The authors provide strong empirical analysis including CKA visualizations.
2. The experiments include many architectures including DINO, ViT, and DeiTs.

**Weaknesses:**

1. The first concern of the paper is that TOAST is extremely sensitive to the hyper-parameters, i.e., which blocks to be replaced. For example, in Table 2 DeiT-S with Linear Approximation, a good choice (10->11) leads to only -0.09% degradation while a bad choice (3->4, 9->11) leads to -7.39% degradation. This suggests that block selection is non-trivial. The paper may be improved with a general theoretical grounded/heuristic guidelines for quickly choosing the optimal layers to replace. In general, based on the results, the last layer may be a good choice, while in Table 3, choosing the last layer seems not to be a good choice as the results are not shown.
2. The second concern of the paper is that since the proposed method is targeting reducing inference efficiency with a few samples, which overlaps strongly with classical compression techniques. However, critical relevant baselines like neural network pruning or quantization are not compared.

**Questions:**

1. In Table 2, DINO-B baseline rows (original) has different accuracy and throughput for identity and linear transformations, is this a typo?

---

> ### Author Response · Authors · 2025-11-21
>
> We thank the reviewer for their thorough assessment and for highlighting the strengths of our work, including the clarity of the presentation, the empirical analyses, and the diversity of architectures evaluated. We appreciate the constructive criticism and offer the following clarifications and planned revisions. We address their questions below:
>
> **W1. Block selection:** We agree with the reviewer that block selection is a non-trivial task. The core idea of TOAST is to provide a simple and efficient method for approximating transformer blocks and reducing computational load. As noted, a key question is which blocks should be skipped, and we fully agree that this is challenging. In our current lightweight approach, as also noted by the reviewer, we select blocks based on downstream task performance, which can vary depending on the architecture. For example, when using DeiT or ViT, it is possible to skip the last blocks while maintaining or even improving performance. While for DiNO it is suggested to skip earlier blocks. Developing conceptual or theory-driven criteria for layer selection is an important future direction, but it is beyond the scope of the current work.
>
> **W2. Baselines:** TOAST is specifically positioned within the post-hoc, training-free paradigm, which targets scenarios where users may not have the resources to fine-tune pre-trained models. Methods such as pruning or quantization often require fine-tuning. Therefore, they operate under a different set of assumptions and have access to a much larger “computational budget” to recover performance. As a result, a direct comparison in a performance table would be computationally unfair and potentially misleading. That said, we are happy to consider including additional baselines or comparisons if the reviewer could suggest specific relevant methods that would be appropriate within our post-hoc, training-free setting.
>
> **Q1. Typo :** We thank the reviewer for catching this inconsistency. This was indeed a typo in the baseline row, and we have corrected it in the revised version of the paper.
>
> We believe our responses address the reviewer’s concerns, and we are happy to provide further clarification if needed.

---

> > ### Comment · Reviewer_sH5D · 2025-11-24
> >
> > I would like to thank the authors for the reply.
> >
> > Regarding W1:
> > Prior work has already shown that intermediate transformer layers exhibit strong feature similarity, and the paper itself confirms this. Given these established similarities, the fact that one block’s output can be approximated by a simple function is not surprising.
> >
> > Thus, the real contribution should be how to systematically choose which blocks to approximate. However, the paper currently relies on manual, hand-picked block pairs, and the results are highly sensitive. Without such method, the contribution feels incomplete.
> >
> > Regarding W2:
> > I appreciate the clarification, but pruning and quantization need not always require fine-tuning. Several compression techniques are training-free, meaning no retraining or weight updates are performed, which puts them in the same “post-hoc, training-free” kind as TOAST. For example:
> >
> > - Raw weight pruning (magnitude-based) — no finetuning.
> > - WANDA:  Simple and Effective Pruning Approach for Large Language Models  (ICLR 2024).
> > - DISP-LLM: Dimension-Independent Structural Pruning for Large Language Models (NeurIPS 2024)
> >
> > Because TOAST is positioned as a general training-free compression method, including at least the strongest training-free pruning or quantization baselines is necessary.

---

> > > ### Author Response · Authors · 2025-12-03
> > >
> > > We thank the reviewer for their thoughtful follow-up questions and constructive feedback. We address the points below:
> > >
> > > **Novelty and use of CKA for block selection:** Prior work has shown that intermediate transformer layers exhibit representational similarity, but, to our knowledge, no existing work has demonstrated that entire transformer blocks can be functionally approximated by a single closed-form linear operator and therefore removed without retraining. This structured, block-level functional replacement differs fundamentally from existing works, and we therefore consider it a valuable contribution.
> > >
> > > We agree that identifying which blocks to skip is itself an important problem and following prior work [1,2,3], we initially adopt CKA as a practical and architecture-agnostic heuristic. However, to provide a better heuristic we formalize the use of a linear approximation error metric for identifying approximable blocks, and conduct an ablation study (Section A.1.3) comparing several metrics (MSE, cosine distance, Euclidean distance, CKA, and the linear approximation error) to determine which best predicts which blocks can be approximated without harming downstream accuracy.
> > >
> > > The newly adopted metric measures how well the representation of a later block can be reconstructed from an earlier one using a closed-form least-squares projection. Because the solution is analytic, the metric is extremely cheap to compute and requires as few as 50 random training samples, making it substantially more efficient and sample-efficient than the alternatives.
> > >
> > > Tables 9 and 10 show that the blocks with the lowest linear approximation error are precisely those that incur minimal accuracy drop when approximated, indicating strong predictive alignment. Across three architectures of varying sizes, this metric consistently achieves the highest mean Precision@5 / Recall@5 (0.60 / 0.60). In contrast, the alternative metrics are far less stable across models: cosine and Euclidean distance behave inconsistently, and CKA, while occasionally competitive, is both more expensive and less robust to architectural differences.
> > >
> > > To sum up, linear approximation error provides a reliable heuristic making it a practical and robust default for selecting which transformer blocks to approximate. We have updated the manuscript accordingly, including pseudocode for the new automatic block-selection algorithm (see section A.1.4) based on user-specified approximation budgets (e.g., skipping up to 𝑁 blocks).
> > >
> > > **W2. Comparison to pruning baselines:** We sincerely thank the reviewer for pointing us to relevant baselines we were not previously aware of. We would like to clarify the following: (1) DISP-LLM requires additional training, and (2) Magnitude pruning and WANDA serve a different goal than TOAST. Pruning methods sparsify the network by zeroing weights to reduce memory footprint, whereas our method removes entire blocks, thereby reducing both memory footprint as well as inference latency and GFLOPs (as reported in the Tables). To the best of our knowledge, pruning does not necessarily guarantee speed-up, since it needs hardwares and kernels that can exploit the sparsity.
> > >
> > > We believe TOAST can indeed be viewed as a structured sparsification method with an additional locality constraint: to remove an entire block, the corresponding weight matrices must be jointly approximated by a single linear operator, effectively enforcing contiguous, block-aligned sparsity. This constraint is significantly stricter than unstructured sparsity and leads directly to real computational gains.
> > >
> > > We appreciate the reviewer’s insights and we have revised the manuscript to (1) clarify our novelty claim, (2) add additional related works, (3) provide the pseudo-algorithm for automatic skip selection based on the new, more efficient, heuristic, and (4) provide an ablation study to support the choice of the heuristic. We believe these revisions will strengthen the presentation of our contribution.
> > >
> > > We thank the reviewer once again for their insightful comments and hope that this response further clarifies our position.
> > >
> > > ---
> > >
> > > [1] S Kornblith, et al. Similarity of neural network representations revisited.
> > >
> > > [2] T. Nguyen, et al. Do wide and deep networks learn the same things? uncovering how neural network representations vary with width and depth.
> > >
> > > [3] S. Venkataramanan, et al. Skip-attention: Improving vision transformers by paying less attention.

---

### Official Review · Reviewer_RS29 · 2025-10-28

**Soundness:** 3
**Presentation:** 3
**Contribution:** 2
**Rating:** 4
**Confidence:** 3

**Summary:**

The paper proposes TOAST, a training-free framework that replaces spans of transformer blocks with lightweight closed-form mappings-either the identity or a single linear map-estimated from a small subset of activations (≈500 images) between a “source” block s and a later “end” block e, motivated by block-wise representational redundancy measured via CKA. TOAST is applied to pre-trained vision transformers (ViT/DeiT/DINOv2) for classification, showing modest accuracy drops (sometimes gains) alongside parameter/GFLOPs reductions and throughput increases; the approach often works best when approximating late blocks, and scales to ViT-L; a small sample ablation suggests performance plateaus by ~500 samples; and a zero-shot OpenCLIP experiment reveals that late-layer approximation can be catastrophic in multimodal settings.

**Strengths:**

1. The paper is well written and easy to follow.

2. Simple, training-free idea with closed-form estimator. Clear formulation with identity/linear translators; easy to implement and deploy.

**Weaknesses:**

1. **Limited experimental scope.** The method is evaluated only on classification tasks, which may rely more heavily on the classification head rather than the transformer backbone itself. Broader evaluation on more complex tasks (e.g., segmentation or detection) is needed to verify whether a single translator is sufficient.

2. **Evaluation protocol may favor TOAST.** Although the appendix includes runs with original heads, these appear only for certain models and datasets, and their reporting format is inconsistent with the main experiments. This inconsistency makes it unclear how performance varies across all configurations. Reporting both protocols-using the original head and retrained head-side by side would better isolate the backbone’s contribution.

3. **Inconsistent layer selection.** The paper does not clearly explain how layers are chosen for reporting. While the focus seems to be on early and late layers, the selections appear somewhat arbitrary, making comparisons difficult. Including more results for middle layers and providing a clear rationale for the chosen spans would improve transparency and interpretability.

4. **Limited interpretability of when and why it works.** CKA-based similarity is used to justify redundancy between layers, but CKA is known to be sensitive to representation scaling and other factors. The observed correlations between similarity heatmaps and downstream performance remain mostly observational. A stronger causal or ablation-based analysis would provide greater confidence in the method’s underlying mechanisms.

**Questions:**

See weaknesses.

---

> ### Author Response · Authors · 2025-11-21
> **[1/2]**
>
> We thank the reviewer for their thorough feedback and for highlighting the strengths of our work, including its clarity, simplicity, and ease of implementation. We appreciate the constructive criticism and offer the following clarifications and planned revisions. We address their questions below:
>
> **W1. Backbone vs Head:**  To isolate the contribution of the TOAST-compressed backbone, we train only a single linear classification head for 5 epochs. This setup ensures that strong performance can be achieved without significant contributions from the classification head, but rather from the backbone itself. Additionally, it is not clear to us why classification should be more prone to relying on the head than other downstream tasks. We would be grateful if the reviewer could elaborate on this point so that we can address it more effectively in the final version.
>
> However, as stated in the paper, evaluating TOAST on additional downstream tasks is indeed an important next step. Thus, we conducted a preliminary experiment on semantic segmentation using the MIT Scene Parsing Benchmark[1], employing the same ViT-S backbone as well as the same setup as for the classification task. 500 samples are used to estimate the transformation and the pre-trained backbone is kept frozen, while only a simple segmentation head is trained. Performance are measured using the Mean Intersection over Union (mIoU) score. Given the time constraint, these results are not intended to reach SOTA performance, but rather to maintain methodological consistency and to demonstrate that TOAST is applicable beyond classification tasks. Although these experiments are not yet exhaustive, they follow the trends observed in our primary results and further support the generality of our approach.
>
> | Approximation       | Identity | Linear | MLP  |
> |---------------------|----------|--------|------|
> | Baseline            | 0.31     | 0.31   | 0.31 |
> | $0 \rightarrow 1$   | 0.18     | 0.27   | 0.27 |
> | $1 \rightarrow 2$   | 0.26     | 0.29   | 0.29 |
> | $2 \rightarrow 3$   | 0.23     | 0.30   | 0.30 |
> | $9 \rightarrow 10$  | 0.29     | 0.29   | 0.29 |
> | $10 \rightarrow 11$ | 0.30     | 0.30   | 0.30 |
> | $1 \rightarrow 3$   | 0.14    | 0.27   | 0.26
> | $2 \rightarrow 5$   | 0.11   | 0.25   | 0.24
> | $9 \rightarrow 11$   | 0.29    | 0.28   | 0.27
>
> These results highlight that it is possible to approximate several blocks (e.g., $10 \rightarrow 11$ or $9\rightarrow 11$) generally maintaining or slightly reducing performance. This suggests that later blocks contain more redundant high-level representations, making them easier to approximate effectively.
>
> Additionally, results show that even in a more complex task like segmentation, a linear transformation is often sufficient to approximate the blocks while preserving most of the model’s performance.
>
> We will provide a complete set of results including other backbones, for all layer combinations and transformation types in the revised manuscript.

---

> ### Author Response · Authors · 2025-11-21
> **[2/2]**
>
> **W2. Original vs trained heads:** We agree with the reviewer and appreciate this suggestion for improving clarity. In our revised manuscript, we have updated Table 12 in the appendix to have the same approximations as the tables in the main paper, as well as a side-by-side comparison with the trained-from-scratch heads allowing for a direct and consistent comparison of results with and without the original head. This clearly shows that the classification head is not the main driver for the final performance, as the same approximations have consistent relative ranking between the pre-trained and newly trained classification heads. We note that this comparison is necessarily limited to DeiT-S and ViT-S on ImageNet-1K, as these are the only configurations where pre-trained classification heads are provided [2,3].
>
> | Encoder | Approximation | Acc. (Original Head) | Acc. (Retrained Head) |
> | :--- | :--- | :--- | :--- |
> | DeiT-S | original | 79.66 | 73.85 $\pm$ 0.39 |
> | DeiT-S | $10 \rightarrow 11$ | 78.78 | 73.78 $\pm$ 0.28 |
> | DeiT-S | $2 \rightarrow 3$ | 78.69 | 73.19 $\pm$ 0.19 |
> | DeiT-S | $3 \rightarrow 4, 9 \rightarrow 10$ | 77.25 | 71.35 $\pm$ 0.22 |
> | DeiT-S | $3 \rightarrow 4, 9 \rightarrow 11$ | 72.44 | 68.39 $\pm$ 0.13 |
> | ViT-S | original | 79.86 | 73.24 $\pm$ 0.13 |
> | ViT-S | $3 \rightarrow 4$ | 78.25 | 71.40 $\pm$ 0.22 |
> | ViT-S | $2 \rightarrow 3$ | 78.25 | 71.26 $\pm$ 0.03 |
> | ViT-S | $4 \rightarrow 5$ | 77.66 | 70.98 $\pm$ 0.16 |
> | ViT-S | $1 \rightarrow 2$ | 76.62 | 70.32 $\pm$ 0.38 |
>
>
> **W3, W4. Block selection and CKA:** The core idea of TOAST is to introduce an easy and efficient method for approximating transformer blocks and reduce the computational load. As the reviewer points out, a key question is which blocks should be skipped, and we totally agree with that. However, in our current lightweight approach, we select blocks based on the resulting downstream task performance. Developing more conceptual or theory-driven criteria for layer selection is an important direction for future work, but it is a non-trivial research question and, therefore, beyond the scope of this work.
>
> Similarly, while CKA and related similarity metrics provide useful observational insights into block redundancy, they are not sufficient to establish causal relationships. In the revised manuscript, we will explicitly acknowledge this limitation and frame the development of stronger, theory-based analyses as a key open problem for future research.
>
> We appreciate the reviewer’s feedback and believe that we have thoroughly addressed the concerns. We’re happy to elaborate further if helpful.
>
> -----
>
> [1]  B. Zhou et al., Scene Parsing through ADE20K Dataset.
>
> [2] Touvron et al., Training data-efficient image transformers & distillation through attention.
>
> [3] Beyer et al., Better plain vit baselines for imagenet-1k

---

> > ### Comment · Reviewer_RS29 · 2025-11-26
> >
> > I appreciate the authors’ detailed rebuttal and the additional experiments, which have addressed some of my concerns.
> >
> > That said, one major issue remains: how to select which blocks to skip. The authors state that they choose blocks based on downstream task performance, but the number of possible block-skip combinations could grow rapidly in larger models. This raises practical concerns about scalability and usability. I understand that deriving a fully theoretical criterion may be challenging, but relying on what appears to be a largely empirical or random search may hinder adoption of the method in real-world settings. Providing at least a more systematic procedure or heuristic for selecting blocks would substantially strengthen the paper.

---

> > > ### Author Response · Authors · 2025-12-03
> > >
> > > We thank the reviewer once again for taking the time to read our rebuttal. We would like to clarify our approach as well as what we stated in the rebuttal.
> > >
> > > TOAST does not rely on exhaustive or random search. Following prior work [1,2,3], we originally used CKA similarity to identify redundant blocks. Specifically: (i) given a model and 100 samples from the dataset, we extract token representations from all layers; (ii) we aggregate tokens within each sample by averaging them, yielding 100 pooled representations per layer; (iii) we compute CKA similarity across all pairs of layer representations to form the similarity matrix shown in Figure 1; (iv) TOAST is then applied to the candidates suggested by this matrix.
> > >
> > > However, we agree with the reviewer that CKA can be sensitive to shifts in latent space and may not generalize uniformly across architectures, as we also mentioned in the related work section [1].
> > >
> > > To address this concern, we performed a new ablation study (Section A.1.3) comparing several metrics including MSE, cosine distance, Euclidean distance, and CKA, to evaluate which best predicts which blocks can be approximated without degrading downstream performance.
> > >
> > > In addition, we now formalize the use of a linear approximation error metric for identifying blocks to approximate, by measuring how well the representation of a later block can be reconstructed from an earlier block using a closed-form least-squares projection followed by a similarity computation in the target latent space. Because the projection has an analytic solution, the metric is extremely cheap to compute and requires as few as 50 random training samples, making it substantially more efficient and sample-efficient than existing alternatives.
> > >
> > > Tables 9 and 10 show that the blocks with the lowest linear approximation error are precisely those that incur minimal accuracy drop when approximated, indicating strong predictive alignment. Across three architectures of varying sizes, this metric consistently achieves the highest mean Precision@5 / Recall@5 (0.60 / 0.60). In contrast, the alternative metrics are far less stable across models: cosine and Euclidean distance behave inconsistently, and CKA, while occasionally competitive, it can be more expensive and less robust to architectural differences.
> > >
> > > To sum up, linear approximation error provides a reliable heuristic making it a practical and robust default for selecting which transformer blocks to approximate. We have updated the manuscript accordingly, including pseudocode for the new automatic block-selection algorithm (see section A.1.4) based on user-specified approximation budgets (e.g., skipping up to 𝑁 blocks).
> > >
> > > We hope this fully addresses the reviewer’s last concern, and we thank them again for their thoughtful and helpful comments.
> > >
> > > ----
> > >
> > > [1] Davari M, et al. "Reliability of cka as a similarity measure in deep learning."

---

### Official Review · Reviewer_NEKG · 2025-11-05

**Soundness:** 3
**Presentation:** 3
**Contribution:** 3
**Rating:** 6
**Confidence:** 2

**Summary:**

This paper proposes Transformer Optimization using Adaptive and Simple Transformations (TOAST) which looks at simple ways to approximate entire transformer blocks in vision transformers. Parts of the model can then be replaced, reducing parameter counts and making the models smaller and more efficient. The method was examined using MNIST, F-MNIST, CIFAR-10, CIFAR-100, and ImageNet 1k. In general, the method has only slight degredation in task performance for DEiT-S and ViT-L with efficiency and speed games, but suffers more for DiNO-B. Overall, this is a potentially interesting paper to the community.

**Strengths:**

A novel method that can make vision transformer models that are already trained smaller and more efficient.

**Weaknesses:**

I'm not as familiar with vision tasks as NLP tasks so it is a bit harder for me to evaluate this. However, my understanding is that MNIST and CIFAR-10 are easier tasks. ImageNet-1K is hard, but is quite old. A potential weakness might be the difficulty of the test sets. However, I'll state again to the other reviewers and area chairs that I'm not as familiar as with other modalities.

**Questions:**

You argue 500 examples are enough and show some plots to justify this. However, many tasks in AI have a long-tailed distribution. Did you look at parts of the longer tails to see if this is the case? For instance, you might only lose a small percentage of absolute performance, but have something that is less robust and only works on the dominant class.

Any idea how this would perform on other tasks? For instance, a text or audio task which also have pre-trained transformer architectures where your method could be applied.

---

> ### Author Response · Authors · 2025-11-21
>
> We thank the reviewer for their positive evaluation and constructive feedback. We address their questions below:
>
> **W1. Choice of datasets:** To justify the applicability of TOAST, we selected both simple datasets, such as MNIST and CIFAR10, as well as more complex ones like CIFAR-100 and ImageNet-1K. These are standard and widely recognized benchmarks for image classification. We also want to emphasize that our study targets accessible deep learning under computational constraints. In this setting, ImageNet-1K remains the standard proxy for accessible and high-performance downstream tasks.
>
> **Q1. Results robustness:** We would like to clarify that the set of 500 samples is used *only* for computing the linear transformation between the two latent spaces to approximate. All reported results use the overall accuracy computed across the entire test set. This follows standard evaluation protocols: models must perform well across the full distribution, and any bias toward dominant classes is naturally penalized by this metric.
>
> **Q2. Transferability to other domains:** We agree with the reviewer that analyzing the applicability of TOAST to other domains is an interesting experiment. Given the time constraint, we performed a preliminary experiment to extend TOAST to text transformers. Specifically, we used ModernBERT-base [1] on the AG News [2] text-classification benchmark. We applied the exact same protocol as in our vision experiments: we used 500 training samples to fit a linear transformation between the blocks to approximate the model with frozen weights, and we trained only a linear classifier on the resulting embeddings for 5 epochs using 3 different seeds. These results are not intended to achieve SOTA performance, but rather to maintain methodological consistency and demonstrate the potential of TOAST and its applicability to other domains beyond vision.
>
> These initial results confirm that TOAST transfers effectively to non-vision domains and is architecture-agnostic, operating consistently across both vision and language models. While these experiments are not yet exhaustive, they align with the trends observed in our primary results and further support the generality of our approach. For example, when approximating 10 blocks ($12 \rightarrow 22$), we can improve efficiency while slightly decreasing the accuracy (87% vs 83%), while in more conservative settings (e.g., $12 \rightarrow 15$) we approximate less blocks while decreasing the accuracy by only 1%. We will include additional results (including all the approximations) in the final manuscript, such as an ablation over different transformations (identity, linear, MLP), efficiency metrics (GFLOPS, throughput), and averages across multiple seeds.
>
> | Approximation       | Mean Accuracy |
> |---------------------|------|
> | Baseline            | 0.87 |
> | $5 \rightarrow 9$   | 0.86 |
> | $12 \rightarrow 15$ | 0.86 |
> | $12 \rightarrow 22$ | 0.83 |
> | $18 \rightarrow 22$ | 0.84 |
> | $19 \rightarrow 22$ | 0.85 |
>
> We will also update our limitations section to reflect that TOAST has now been validated preliminarily on text and segmentation tasks, while emphasizing that a comprehensive study across modalities and tasks remains an important direction for future work.
>
> We appreciate the reviewer’s feedback and believe that we have thoroughly addressed the concerns. We’re happy to elaborate further if helpful.
>
> ----
>
> [1] Warner, Benjamin, et al. Smarter, better, faster, longer: A modern bidirectional encoder for fast, memory efficient, and long context finetuning and inference.
>
> [2] Xiang Zhang et al., Character-level Convolutional Networks for Text Classification.

---

### Author Response · Authors · 2025-12-03
**Summary for the New Area Chair**

Dear Area Chair,

Given the unusual circumstances, we would like to provide a brief summary of the review discussion and our rebuttal.

We appreciate that the reviewers found our method to be **novel** (```NEKG```), **simple and easy to follow** (```sH5D```, ```RS29```), and aligned with a **promising and valuable line of research** (```dwJk```). We are also encouraged that they regarded our formulation as **clear** (```sH5D, RS29```) and our approach as **straightforward to implement and deploy** (```RS29```). In addition, we are grateful for the recognition of our **strong empirical analysis** and **extensive experimental design** (```sH5D```).

The main concerns were focused on (1) the metric used to select which blocks to approximate and (2) the generalizability of our approach to additional downstream tasks and domains. We took this feedback seriously and conducted substantial new experiments to address each point:

**1. Block Selection Heuristic** (```RS29```, ```sH5D```). Our Actions:
* We add a **robust heuristic** for **automatically selecting** which blocks to approximate. This **computationally lightweight** metric measures how well a later block's representation can be reconstructed from an earlier one using a closed-form least-squares projection and using just 50 samples, without online inference or any other additional training.
*   To improve clarity on the selection strategy , we include **detailed pseudo-code** explaining the block selection process under user-specified approximation budget (Algorithms 1 and 2 in Appendix A.1.4);
*   We perform an **expanded ablation study** (Table 10 in Appendix A.1.3) to justify the use of the new heuristic as well as prove his reliability.
*   We **integrated** the new heuristic description into Section 3 of the main paper, qualitative results into Section 4.1 as well as qualitative results in Table 9 in Appendix.

**2. Applicability to the Text Domain** (```NEKG```, ```RS29```, ```dwJk```). Our Action:
*   We added **text classification results** on the AG News dataset using ModernBERT-B as encoder and the *same simple setup* as in the vision experiments. Results provided in Table 5 in the main paper and Table 17 in the Appendix, empirically demonstrate that TOAST generalizes beyond the vision domain.

**3. Applicability to Additional Downstream Tasks** (```RS29```). Our Action:
*   We provide **semantic segmentation results** on the MIT Scene Parsing Benchmark dataset using ViT-S and DiNO-B and the *same simple setup* as in the other experiments, except for the head that is, in this case, a segmentation head. Results in Table 6, empirically demonstrate that TOAST applies effectively to more complex vision tasks.

We thank the AC for their time and careful consideration, as well as the reviewers for their efforts in helping us substantially improve the paper. We believe the revised paper is significantly stronger and hope this summary is helpful.

Sincerely,

The Authors

---

### Meta-Review · Area_Chair_n8FW · 2026-01-07

**Summary:**

The paper proposes TOAST, a training-free method to reduce Transformer complexity by replacing blocks with linear transformations or identity mappings. While the reviewers acknowledge the method's simplicity and the practical appeal of a training-free solution, the overall consensus remains below the acceptance threshold. The primary criticisms center on limited technical novelty and insufficient empirical contextualization. Reviewers expressed concerns that the observed representational similarity is a well-known phenomenon and that replacing blocks with linear layers is an incremental contribution. Furthermore, the lack of direct comparisons with state-of-the-art training-free pruning and quantization methods makes it difficult to assess if TOAST offers any real advantage over established compression paradigms.

**Reviewer Concerns:**

Addressed by Rebuttal:

- Scope Extension: The authors added preliminary results on NLP tasks and semantic segmentation, showing that the concept of block-redundancy can be observed beyond vision classification.

- Selection Heuristic: A "Linear Approximation Error" metric was introduced to automate the selection of blocks to be replaced, partially addressing the concern that the original results were based on manual cherry pick.

Outstanding Part:

- Lack of Competitive Baselines (sH5D, dwJk): A major point of contention is the absence of comparisons with SOTA training-free pruning (like WANDA or Magnitude pruning). The authors argued these are "different paradigms (structure/non-structure)," but since both target inference efficiency without retraining, the lack of a head-to-head comparison is a significant gap. Without this, the community cannot judge if TOAST is a superior alternative or merely a less effective version of layer-level pruning.

- Incremental Novelty (dwJk, sH5D): Reviewers noted that replacing a block with an identity or a single linear layer is conceptually equivalent to structured pruning or depth-wise stitching, both of which are well-studied. The "insight" that layers are similar is already widely documented (e.g., CKA studies), and the proposed solution is viewed as a straightforward application of these known properties rather than a novel scientific breakthrough.

- Sensitivity and Performance Trade-offs (sH5D, dwJk): The accuracy drops are often non-negligible, especially on smaller models where efficiency is most critical. On models like DINO-B, the degradation was significant.

**Reviewer Scores:**

Reviewer NEKG (Score: 6): Likely to remain a 6. This reviewer was the least critical and focused on the novelty of the vision transformer application, though their confidence was low (2/5).

Reviewer RS29 (Score: 4): Likely to stay at 4. Although the authors provided a heuristic for layer selection, the reviewer’s broader point about the method being "observational" rather than "causal" remains unaddressed. The heuristic is seen as a post-hoc patch rather than a principled design.

Reviewer sH5D (Score: 4): Likely to stay at 4. This reviewer was very firm about the missing baselines (WANDA, etc.). The authors' refusal to include these in a quantitative comparison is likely viewed as a failure to meet the rigor expected.

Reviewer dwJk (Score: 4): Likely to stay at 4. Their concern about the "practical utility" and "limited impact" remains, as the parameter reduction (often <10%) is often not worth the corresponding accuracy loss when compared to what standard pruning can achieve.

---

### Decision · Program_Chairs · 2026-01-26

Reject